# PAINET: A Principled Efficient Transformer for 3D Dynamics Modeling

**Kai Yang**[1,2], **Yuqi Huang**[3], **Junheng Tao**[5,6], **Wanyu Wang**[4], **Qitian Wu**[7*]

[1]Department of Computer Science and Engineering, Shanghai Jiao Tong University
[2]Shanghai Artificial Intelligence Laboratory
[3]School of Artificial Intelligence, Shanghai Jiao Tong University
[4]SJTU Paris Elite Institute of Technology, Shanghai Jiao Tong University
[5]Department of Physics, Harvard University
[6]Harvard-MIT Center for Ultracold Atoms
[7]Eric and Wendy Schmidt Center, Broad Institute of MIT and Harvard
{icarus1411, hhhuangq, wwyspeit}@sjtu.edu.cn
jtao@fas.harvard.edu, wuqitian@mit.edu

## Abstract

Modeling 3D dynamics is a fundamental problem in multi-body systems across scientific and engineering domains and has important practical implications in object trajectory prediction and simulation. While recent GNN-based approaches have achieved strong performance by enforcing geometric symmetries, encoding high-order features or incorporating neural-ODE mechanics, they typically depend on explicitly observed structures and inherently fail to capture the unobserved interactions that are crucial to complex physical behaviors and dynamics mechanism. In this paper, we propose PAINET, a principled SE(3)-equivariant transformer for learning all-pair interactions in multi-body systems. The model comprises: (1) a novel physics-inspired attention network derived from the minimization trajectory of an energy function, and (2) a parallel decoder that preserves equivariance while enabling efficient inference. Empirical results on diverse real-world benchmarks, including human motion capture, molecular dynamics, and large-scale protein simulations, show that PAINET consistently outperforms recently proposed models, yielding 4.7% to 41.5% error reductions in 3D dynamics prediction with comparable computation costs in terms of time and memory. Our codes, baseline models and datasets are available at https://github.com/Icarus1411/PAINET.

## 1 Introduction

Modeling the 3D dynamics is a fundamental challenge across a wide spectrum of scientific and engineering disciplines, encompassing molecular dynamics (Schütt et al., 2017; Unke et al., 2021), celestial mechanics (Sanchez-Gonzalez et al., 2019), physical simulation (Battaglia et al., 2016), etc. Traditional approaches are grounded in classical mechanics, leveraging physical laws such as Newtonian dynamics or force-field models. Despite decent accuracy and interpretability, these methods require substantial computational costs for solving partial differentiable equations in systems with complex interactions or large numbers of particles (Kipf et al., 2018).

Recent years have witnessed the promise of data-driven approaches, particularly those based on deep learning methods. Graph neural networks (GNNs), which treat particles as nodes and their interactions as edges, have emerged as a powerful paradigm for learning physical dynamics directly from observed data (Battaglia et al., 2016; Sanchez-Gonzalez et al., 2019; Pfaff et al., 2021). A typical GNN model is Equivariant Graph Neural Network (EGNN) (Satorras et al., 2021) which has shown strong capability for modeling N-body dynamics by preserving SE(3)-equivariance, a property recognized as an essential geometric prior in physical systems (Bronstein et al., 2017). It inspires a series of follow-up extensions with higher-order features (Liu et al., 2022; Cen et al.,

---
*Corresponding author.

2024), space transformation (Du et al., 2022), equivariant local frames (Yin et al., 2025), local attention (Fuchs et al., 2020), and temporal convolution (Xu et al., 2024), which pushes the frontier of state-of-the-art performance on diverse benchmarks.

However, current models predominantly hinge on explicitly observed structures, e.g., an adjacency graph extracted from distances of particles, for computing representations. This design neglects unobserved all-pair interactions among particles, which play an indispensable role in the dynamics mechanism. For example, in atomic and molecular systems, the long-range tails of the Van der Waals-type potentials (London, 1937) are often overshadowed by the stronger short-range interactions (French et al., 2010), yet neglecting them degrades long-horizon trajectory accuracy and obscures long-range correlations (Sagui & Darden, 1999; Defenu et al., 2023). Another example is the spontaneous formation of unobserved structures, such as in crystallization (Schreiber & Fersht, 1996) and protein folding (Richards, 1977; Cao, 2020). In these cases, the observed structures represent only a transient temporal snapshot; stipulating the model on fixed observed structures may lead to systematic bias and significant error accumulation in time, as the true interaction landscape dynamically evolves (Taudt et al., 2015).

Despite the significance, modeling unobserved interactions is non-trivial since the searching space for latent structures (which involves the potential interactions between arbitrary pairs of particles) is exponential to the number of particles and the optimal structures can dynamically evolve across time. Without a principled formulation for unobserved interactions and how they couple with the dynamics mechanism, uncovering the latent structures is underconstrained, computationally prohibitive, and also prone to spurious correlations. Furthermore, it is not straightforward to preserve the SE(3)-equivariance property for models accommodating the latent interactions beyond observed structures.

In this paper, we propose PAINET (**P**hysics-inspired **A**ll-pair **I**nteractions **Net**work) for 3D dynamics modeling. Starting from an energy function that quantifies the latent interactions among particles through smoothness criteria of particle embeddings, we derive a principled attention mechanism with adaptive pairwise mappings that captures long-range, particle-type-specific dependencies. Moreover, the model is implemented with a parallel decoder that preserves SE(3)-equivariance while maintaining efficiency in inference.

We evaluate PAINET on diverse real-world datasets, including human motion sequences, molecular dynamics (MD17), and large-scale protein simulations. Empirical results show that PAINET consistently outperforms recently proposed methods with up to 41.5% / 18.0% / 4.7% decrease of mean square errors (MSEs) in motion capture / MD17 / protein dynamics simulation, while consuming comparable computation costs as those competing models. Furthermore, we conduct comprehensive ablation studies that corroborate the efficacy of the key proposed components and demonstrate that the computation costs of PAINET grow nearly linearly w.r.t. particle numbers and time steps.

**The contributions of our work are summarized below:**

▶ **An energy-based formulation for latent structure learning in 3D dynamics.** We formulate the problem of uncovering latent structures in 3D dynamics as minimization of an energy function that characterizes the smoothness of particle embeddings in latent space, which paves the way to a principled approach for accommodating the unobserved interactions into 3D dynamics modeling.

▶ **A physics-inspired attention network with SE(3)-equivariance and efficient decoding.** On top of the formulation, we propose a new attention network with adaptive pairwise mappings to capture long-range, particle-type-specific dependencies beyond observed structures at each step. The model is compatible for parallel decoding that preserves SE(3)-equivariance with efficient inference.

▶ **Comprehensive empirical evaluation and comparison in multi-faceted aspects.** As demonstrated by experiments on eleven datasets ranging from human motions to molecules and proteins, the proposed model achieves 4.7%-41.5% improvements in terms of 3D dynamics prediction as measured by MSE and shows desired scalability to large-scale multi-body systems with increasing numbers of particles and simulation time steps.

## 2 PRELIMINARIES

**Notations and Problem Formulation.** The 3D dynamics modeling can be formulated as learning over a sequence of geometric graphs $\mathcal{G}^{(t)} = (\mathcal{V}, \mathcal{E}, \mathbf{F}, \mathbf{A}, \mathbf{X}^{(t)}, \mathbf{V}^{(t)})$, where $\mathcal{V}$ denotes the set of particles (nodes) and $\mathcal{E}$ represents the set of observed interactions (edges). $\mathbf{F} \in \mathbb{R}^{N \times d_f}$ denotes the

node features and $\mathbf{A} \in \mathbb{R}^{N \times N \times d_e}$ is an edge-attribute tensor, where $\mathbf{a}_{ij} \in \mathbb{R}^{d_e}$ denotes the feature vector associated with the edge between nodes $i$ and $j$. $\mathbf{X}^{(t)} = [\mathbf{x}_1^{(t)}; \ldots; \mathbf{x}_N^{(t)}] \in \mathbb{R}^{N \times 3}$ and $\mathbf{V}^{(t)} = [\mathbf{v}_1^{(t)}; \ldots; \mathbf{v}_N^{(t)}] \in \mathbb{R}^{N \times 3}$ denote the position and velocity matrices at time step $t$, respectively, where $\mathbf{x}_i^{(t)}$ and $\mathbf{v}_i^{(t)}$ denote the position and velocity of node $i$ in the 3D coordinate system. The problem can be described as: given the initial observation $\mathcal{G}^{(0)} = (\mathcal{V}, \mathcal{E}, \mathbf{F}, \mathbf{A}, \mathbf{X}^{(0)}, \mathbf{V}^{(0)})$, the model aims at predicting the future dynamics trajectory $\mathbf{X}^{(1:T)} = \{\mathbf{X}^{(1)}, \ldots, \mathbf{X}^{(T)}\}$.

**Equivariance.** The notion of equivariance formalizes a type of symmetric property in geometric space regardless of the chosen coordinate system. Formally, let $g \in G$ denote an abstract group, then a function $\mu : \mathcal{X} \to \mathcal{Y}$ is defined as equivariant with respect to $g$ if there exists $S_g : \mathcal{Y} \to \mathcal{Y}$ for the set of transformations $T_g : \mathcal{X} \to \mathcal{X}$ on $g$ such that: $\mu(T_g(\mathbf{x})) = S_g(\mu(\mathbf{x}))$. In this paper, we focus on three types of equivariance in special Euclidean group SE(3), including rotations, translations, and permutations. For 3D dynamics, we can instantiate $T_g$ and $S_g$ as translation $\mathbf{g} \in \mathbb{R}^3$ and orthogonal rotation matrix $\mathbf{Q} \in \mathrm{SO}(3)$, respectively, where $\mathrm{SO}(3)$ is special Orthogonal group in 3D space. Then, an SE(3)-equivariant predictor $f$ for next-step positions satisfies:

$$f\big(\mathbf{Q}\mathbf{X}^{(t)} + \mathbf{1}\,\mathbf{g}^\top\big) = \mathbf{Q}\,f(\mathbf{X}^{(t)}) + \mathbf{1}\,\mathbf{g}^\top, \quad \mathbf{1} = [1,1,1]^\top. \tag{1}$$

We enforce this property throughout the architecture as an inductive bias for plausible prediction. Further details on translation, rotation and permutation properties are presented in Appendix B.

**Equivariant Graph Neural Networks.** Standard GNNs are permutation-equivariant with respect to node relabeling. Beyond this, Equivariant Graph Neural Network (EGNN) (Satorras et al., 2021) attains SE(3)-equivariance via the message passing layers that operate on invariant quantities (e.g., pairwise distances and relative positions). A typical EGNN layer updates node embeddings $\mathbf{h}_i \in \mathbb{R}^d$ and positions $\mathbf{x}_i \in \mathbb{R}^3$ from layer $l$ to $l + 1$ via:

$$
\begin{aligned}
\mathbf{m}_{ij}^{(l)} &= \phi_m(\mathbf{h}_i^{(l)}, \mathbf{h}_j^{(l)}, \|\mathbf{x}_i^{(l)} - \mathbf{x}_j^{(l)}\|^2, \mathbf{a}_{ij}), \\
\mathbf{h}_i^{(l+1)} &= \phi_h\left(\mathbf{h}_i^{(l)}, \sum_{j \in \mathcal{N}(i)} \mathbf{m}_{ij}^{(l)}\right), \\
\mathbf{x}_i^{(l+1)} &= \mathbf{x}_i^{(l)} + \sum_{j \in \mathcal{N}(i)} (\mathbf{x}_i^{(l)} - \mathbf{x}_j^{(l)}) \cdot \phi_x(\mathbf{m}_{ij}^{(l)}),
\end{aligned}
\tag{2}
$$

where $\phi_m$, $\phi_h$ and $\phi_x$ are commonly instantiated as MLPs, and $\mathcal{N}(i)$ denotes the set of neighbored nodes of node $i$. Built upon the main design of EGNN, a recent model EGNO (Xu et al., 2024) additionally incorporates temporal convolution to capture temporal correlations, while a follow-up work HEGNN (Cen et al., 2024) augments EGNN with higher-order interactions to enhance the model's expressivity. Furthermore, GF-NODE (Sun et al., 2025) further adopts a neural-ODE formulation for accommodating continuous-time evolution on graphs. Despite their effectiveness for modeling the spatial-temporal patterns hinging on observed structures, these models restrict message passing at each layer to local neighborhoods stipulated by $\mathcal{E}$ and overlook the latent interactions that are unobserved yet make large differences in dynamics mechanism.

## 3 LEARNING ALL-PAIR INTERACTIONS FOR 3D DYNAMICS

In this section, we propose PAINET (*Physics-inspired All-pair Interactions Network*) for 3D dynamics modeling. We first derive a physics-inspired feed-forward layer from the minimization of a regularized energy (Sec. 3.1). We further introduce the model architecture including a principled attentive encoder as well as a parallel equivariant decoder that allows efficient inference (Sec. 3.2). Finally, we present the training and inference details for 3D dynamics prediction (Sec. 3.3).

### 3.1 MODEL FORMULATION

**Energy with Latent Structures.** Learning unobserved interactions among particles is non-trivial, as the searching space goes to the exponential order w.r.t. the number of particles. To cast the problem into a solvable formulation, we start from an energy function that characterizes the plausible latent structures through regularizing the smoothness of particle embeddings in latent space:

$$E(\mathbf{H}, t; \{\rho_{ij}\}) = \sum_i \|\mathbf{h}_i - \mathbf{h}_i^{(t)}\|_2^2 + \lambda \sum_{i,j} \rho_{ij}(\|\mathbf{h}_i - \mathbf{h}_j\|_2^2), \tag{3}$$

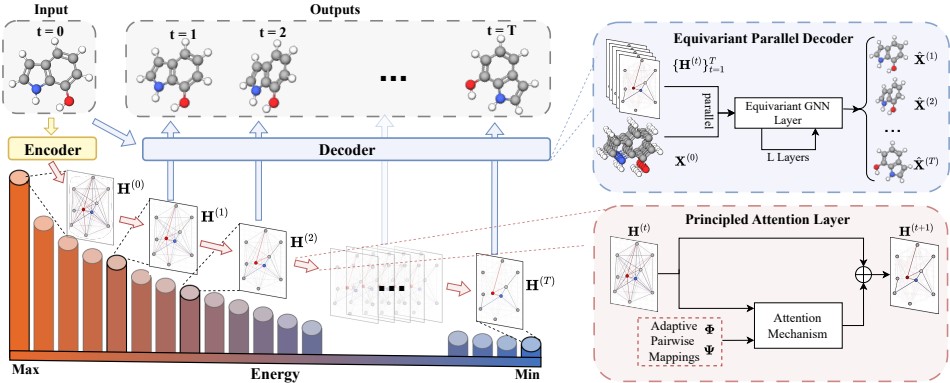

Figure 1: Illustration of PAINET framework. The model takes the initial state (including positions, velocities and observed features such as edge attributes of particles) as input and encode observed information into particle embeddings in latent space. The particle embeddings are updated through a stack of principled attention layers, where each layer corresponds to a descent step on the energy. The attention network includes adaptive pairwise mappings to capture long-range, particle-type-specific dependencies. For decoding, the model harnesses equivariant GNNs that incorporates the observed structural information without breaking SE(3)-equivariance and generates predicted trajectory of particles at multiple time steps in parallel.

where $\mathbf{h}_i^{(t)}$ denotes the current embedding of particle $i$ at layer $t$, $\mathbf{h}_i$ represents its updated (to-be-optimized) embedding for the next layer, and $\mathbf{H} = [\mathbf{h}_1^\top; \dots; \mathbf{h}_N^\top]$ is the stack of the embeddings of $N = |\mathcal{V}|$ particles. Without loss of generality, $\rho_{ij} : \mathbb{R}^+ \to \mathbb{R}$ is a non-linear non-decreasing function specific to particle pair $(i, j)$ and $\lambda > 0$ is a trading weight. Eqn. 3 which extends the quadratic energy introduced by the prior work (Zhou et al., 2004) integrates the local and global smoothness criteria: the first term regularizes the distance between the updated embeddings and the current embeddings, preventing the abrupt change of embeddings between adjacent time steps; the second term penalizes the distance among updated embeddings of all particles, encouraging the global smoothness stipulated by latent structures. The latter essentially accommodates the unobserved interactions in an implicit manner: any two particles whose embeddings are close (resp. distant) in latent space add small (resp. large) quantities to the energy. In this sense, Eqn. 3 characterizes the internal consistency of particle embeddings of any given layer. To improve its robustness, it is natural to assume the concavity of $\rho_{ij}$ that avoids over-regularization on large differences (Yang et al., 2021) that is beneficial for filtering noisy interactions.

**Feed-Forward Layers from Energy Minimization.** In common physical systems, the evolution of particle states often transitions from high-energy to low-energy regimes and ultimately reaches certain equilibrium. Projected to our context, where our goal is to learn desired particle representations that accommodate the unobserved interactions, descending the energy defined by Eqn. 3 corresponds to pursuing a final state that optimizes the internal consistency among particle embeddings in latent space. From this physics-inspired standpoint, we next derive a feed-forward layer that updates particle embeddings whose formed trajectory in latent space minimizes Eqn. 3. To this end, we provide the following result that suggests a closed-form solution for the optimal embedding trajectory where each step (i.e. feed-forward layer) contributes to a rigorous descent step on the non-convex energy.

**Theorem 1** *For any energy function defined by Eqn. 3 with a given $\lambda > 0$, there exists $0 < \eta < 1$ such that the iterative updating rule (from the initial state $\mathbf{h}_i^{(0)}$)*

$$\mathbf{h}_i^{(t+1)} = (1 - \eta)\mathbf{h}_i^{(t)} + \eta \sum_j \frac{\omega_{ij}^{(t)}}{\sum_m \omega_{im}^{(t)}} \cdot \mathbf{h}_j^{(t)}, \quad where \ \omega_{ij}^{(0)} = \left.\frac{\partial \rho_{ij}(h^2)}{\partial h^2}\right|_{h^2 = \|\mathbf{h}_i^{(t)} - \mathbf{h}_j^{(t)}\|_2^2}, \quad (4)$$

*yields a descent step on the energy, i.e., $E(\mathbf{H}^{(t+1)}, t+1; \{\rho_{ij}\}) \leq E(\mathbf{H}^{(t)}, t; \{\rho_{ij}\})$ for any $t \geq 1$.*

The proof follows the principles of convex analysis and Fenchel duality (Rockafellar, 1970). This result implies a principled attentive feed-forward layer where the attention of particle pair $(i, j)$ is

given by the gradient of the pairwise penalty function $\rho_{ij}$ evaluated at the current step. Assuming that $f_{ij}$ denotes the first-order derivative of $\phi_{ij}$, we have the updating rule of one feed-forward layer:

$$\mathbf{h}_i^{(t+1)} = (1-\eta)\mathbf{h}_i^{(t)} + \eta \sum_j \frac{f_{ij}(\|\mathbf{h}_i - \mathbf{h}_j\|_2^2)}{\sum_m f_{im}(\|\mathbf{h}_i - \mathbf{h}_m\|_2^2)} \cdot \mathbf{h}_j^{(t)}, \tag{5}$$

where $f_{ij}$ is a non-negative, decreasing function w.r.t. $h^2 = \|\mathbf{h}_i - \mathbf{h}_j\|_2^2$ such that $\rho_{ij}$ satisfies the concavity and non-negativity.

**Physics-Inspired Attention Mechanism.** The instantiations of $f_{ij}$ have much flexibility. Here, we adopt a concise yet generic polynomial potential form that respects the parity symmetry of $h$. In modern physics, the Landau-Ginzburg potential energy form (Landau et al., 1937) plays a central role in the phenomenological explanation of the formation of matter states ranging from super-conducting (Ginzburg et al., 2009) and ferromagnetic materials (Landau et al., 1937) to complex biochemical media (Hohenberg & Krekhov, 2015). Moreover, it has inspired the Anderson-Higgs mechanism for spontaneous symmetry breaking in quantum field theory (Anderson, 1963; Higgs, 1964). As per this physics motivation, we instantiate $\rho_{ij}$ as a quadratic potential form:

$$\rho_{ij}(h^2) = a_{ij}h^2 - b_{ij}h^4, \quad f_{ij}(h^2) = a_{ij} - 2b_{ij}h^2, \tag{6}$$

where $a_{ij} > 8b_{ij}$ and $b_{ij} > 0$ such that the non-negativity and decreasing property of $f_{ij}$ are guaranteed. Furthermore, we assume particle embeddings are normalized into a unit sphere, i.e., $\|\mathbf{h}_i\|_2 = 1$, so that $\|\mathbf{h}_i - \mathbf{h}_j\|_2^2 = 2 - 2\mathbf{h}_i^\top \mathbf{h}_j$. This property can be easily satisfied in practice by using layer normalization at each layer. Therefore, we have the updating rule of an attention layer:

$$\mathbf{h}_i^{(t+1)} = (1-\eta)\mathbf{h}_i^{(t)} + \eta \sum_j \frac{\phi_{ij} + \psi_{ij}(\tilde{\mathbf{h}}_i^{(t)})^\top \tilde{\mathbf{h}}_j^{(t)}}{\sum_m \phi_{im} + \psi_{im}(\tilde{\mathbf{h}}_i^{(t)})^\top \tilde{\mathbf{h}}_m^{(t)}} \cdot \mathbf{h}_j^{(l)}, \quad \text{where } \tilde{\mathbf{h}}_i^{(t)} = \frac{\mathbf{h}_i^{(t)}}{\|\mathbf{h}_i^{(t)}\|_2}. \tag{7}$$

Here we replace $a_{ij}, b_{ij}$ in Eqn. 6 with variables $\phi_{ij}, \psi_{ij}$ satisfying $\phi_{ij} > \psi_{ij} > 0$. They give rise to adaptive linear mappings for specific particle pairs which is beneficial for capturing informative unobserved interactions specific to particle types and states.

## 3.2 MODEL ARCHITECTURE

Based on the above model formulation, we next present the detailed architecture of the proposed model PAINET for 3D dynamics modeling. Fig. 1 provides an overview of our model framework.

**Principled Attention Network.** We extend the attention layer derived in Sec. 3.1 to parameterized neural networks and present the matrix-form updating rule as implemented with modern deep learning library. For particle embeddings at layer $t$ denoted by $\mathbf{H}^{(t)} \in \mathbb{R}^{N \times d}$, we first compute its key, query and value matrices: $\mathbf{Q}^{(t)} = \mathbf{W}_Q^{(t)}\mathbf{H}^{(t)}, \mathbf{K}^{(t)} = \mathbf{W}_K^{(t)}\mathbf{H}^{(t)}, \mathbf{V}^{(t)} = \mathbf{W}_V^{(t)}\mathbf{H}^{(t)}$, where $\mathbf{W}_Q^{(t)}, \mathbf{W}_K^{(t)}, \mathbf{W}_V^{(t)} \in \mathbb{R}^{d \times d}$ are trainable weights and $d$ is the hidden size. We assume $\tilde{\mathbf{Q}}^{(t)} \in \mathbb{R}^{N \times d}$ and $\tilde{\mathbf{K}}^{(t)} \in \mathbb{R}^{N \times d}$ as the L2-normalized versions of $\mathbf{Q}^{(t)}$ and $\mathbf{K}^{(t)}$, respectively. Then following Eqn. 7, the equivalent matrix-form attention-based updates at layer $t$ can be written as:

$$\begin{aligned}
\mathbf{D}^{(t+1)} &= \text{diag}^{-1}\left(\mathbf{\Phi}\mathbf{1} + (\mathbf{\Psi} \odot (\tilde{\mathbf{Q}}^{(t)}(\tilde{\mathbf{K}}^{(t)})^T))\mathbf{1}\right), \\
\mathbf{H}^{(t+1)} &= (1-\eta)\mathbf{H}^{(t)} + \eta\mathbf{D}^{(t+1)}\left(\mathbf{\Phi}\mathbf{V}^{(t)} + (\mathbf{\Psi} \odot (\tilde{\mathbf{Q}}^{(t)}(\tilde{\mathbf{K}}^{(t)})^T))\mathbf{V}^{(t)}\right),
\end{aligned} \tag{8}$$

where the operator $\text{diag}(\cdot)$ converts the input vector into a diagonal matrix, $\mathbf{1} \in \mathbb{R}^{N \times 1}$ is an all-one vector, $\odot$ denotes the Hadamard product, $\eta \in (0, 1)$ is a hyper-parameter, and $\mathbf{\Phi} = [\phi_{ij}]_{i,j \in \mathcal{V}}$ and $\mathbf{\Psi} = [\psi_{ij}]_{i,j \in \mathcal{V}}$ serve for adaptive pairwise mappings. The model described so far treats 'layers' as equivalent to 'time steps', in which case at each time step the embeddings are updated by one attention layer as defined by Eqn. 8. One can extend the model with multiple attention layers per time step, yet as validated in our experiments this simpler architecture using one attention layer per time step yields the optimal performance (Sec. 4.5).

**Adaptive Pairwise Mappings.** Motivated by the fact that Van der Waals–type potentials employ different coefficients across particle types (e.g., carbon-carbon vs. carbon-hydrogen) (London, 1937), we specify $\mathbf{\Phi}$ and $\mathbf{\Psi}$ to parameterize particle-type-dependent interactions. Assuming the number of particle types (e.g., atomic types for molecules) is $E$, we let $\mathbf{z}_i \in \{0, 1\}^E$ be a one-hot

vector indicating the type of particle $i$ (with $\mathbf{z}_{i,k} = 1$ iff the type index is $k$) and stack them as $\mathbf{Z} = [\mathbf{z}_1^\top; \ldots ; \mathbf{z}_N^\top] \in \{0, 1\}^{N \times E}$. On top of these, we introduce two learnable particle-type–specific lookup embeddings $\mathbf{E}_\phi \in \mathbb{R}^{E \times E}$ and $\mathbf{E}_\psi \in \mathbb{R}^{E \times E}$. The pairwise weight matrices are then calculated by:

$$\boldsymbol{\Phi} = s_1 \cdot \sigma(\mathbf{Z}\mathbf{E}_\phi \mathbf{Z}^\top), \quad \boldsymbol{\Psi} = s_2 \cdot \sigma(\mathbf{Z}\mathbf{E}_\psi \mathbf{Z}^\top), \tag{9}$$

where $\sigma(\cdot)$ is specified as `sigmoid` and $s_1, s_2 > 0$ are two learnable scalars, which ensures the non-negativity of $\phi_{ij}$ and $\psi_{ij}$.

**Parallel Equivariant Decoder.** After the particle embeddings $\{\mathbf{H}^{(t)}\}_{t=1}^T$ are obtained through recurrently operating $T$ attention layers across time steps, we integrate them with the observed graph structures through a parallel decoding network. For each time step $t$, the decoder generates the predicted positions $\widehat{\mathbf{X}}^{(t)}$ from the initial positions $\mathbf{X}^{(0)}$ and velocities $\mathbf{V}^{(0)}$ via message passing of the current particle embedding $\mathbf{H}^{(t)}$ over the observed graph structure $\mathbf{A}$. The predictions for $T$ time steps can be achieved in parallel:

$$\widehat{\mathbf{X}}^{(t)} = \texttt{decoder}(\mathbf{H}^{(t)}, \mathbf{X}^{(0)}, \mathbf{V}^{(0)}, \mathbf{A}), \quad 1 \le t \le T, \tag{10}$$

where we instantiate `decoder` as EGNN defined by Eqn. 2. The full feed-forward flow of our model is described in Alg. 1. Prior works have shown the importance of SE(3)-equivariance for physical plausibility in 3D dynamics modeling (Satorras et al., 2021; Xu et al., 2024). We can show that our model maintains SE(3)-equivariance throughout the encoding and decoding process. As a proof sketch, the embedding $\mathbf{h}_i$ remains invariant to rigid transformations of the input coordinates, while the position $\mathbf{x}_i$ remains equivariant after EGNN. As a result, the model preserves the desired transformation properties under any rotation $\mathbf{Q} \in \mathrm{SO}(3)$ and translation $\mathbf{g} \in \mathbb{R}^3$. The complete proof is deferred to Appendix E.

## 3.3 TRAINING AND INFERENCE

**Training Objective.** With initial positions $\mathbf{X}^{(0)}$ and (optionally) velocities $\mathbf{V}^{(0)}$, the initial particle embeddings are given by an MLP: $\mathbf{H}^{(0)} = \mathrm{MLP}(\mathbf{X}^{(0)}, \mathbf{V}^{(0)})$. The model is trained in a supervised manner to predict future trajectories $\{\mathbf{X}^{(t)}\}_{t=1}^T$. Let $\widehat{\mathbf{X}}^{(t)}$ denote the predicted positions at time step $t$ and $\widehat{\mathbf{x}}_i^{(t)}$ is the predicted position of particle $i$. The training loss is the averaged mean squared error:

$$\mathcal{L}_{\text{traj}} = \sum_{t=1}^T \sum_{i=1}^N \left\| \widehat{\mathbf{x}}_i^{(t)} - \mathbf{x}_i^{(t)} \right\|_2^2. \tag{11}$$

**Inference.** At inference time, PAINET first computes a sequence of particle embeddings $\{\mathbf{H}^{(t)}\}_{t=1}^T$ via recurrently applying the attention network for $T$ steps. These embeddings are then used to generate the predicted positions $\{\mathbf{X}^{(t)}\}_{t=1}^T$ in parallel by the decoder, as shown in Fig. 1 and specifically Alg. 1. This recurrent–parallel framework accommodates all-pair interactions in encoding phase and enables efficient generation that preserves SE(3)-equivariance in the decoding phase.

## 4 EXPERIMENTS

In this section, we apply PAINET to various real-world datasets that involve 3D dynamics modeling in different domains to validate the practical efficacy of the proposed model.

**Evaluation Protocols.** Following Xu et al. (2024), we conduct comparisons on two tasks, referred to as *state to state* (S2S) and *state to trajectory* (S2T). The task of S2S aims to directly predict the final state ($T = 1$) and assesses the Mean Square Error (MSE) from the ground truth, referred to as Final Mean Squared Error (F-MSE). The task of S2T aims to predict the trajectories including multiple designated future time steps ($t = 1, \cdots, T$ where $T > 1$), as measured by MSE averaged across time steps, referred to as Average Mean Squared Error (A-MSE) (see Appendix F.1 for details).

**Competitors.** We compare PAINET with representative classic models including Linear Dynamics (Linear) (Satorras et al., 2021), Radial Field Networks (RF) (Köhler et al., 2019), Message Passing Neural Networks (MPNN) (Gilmer et al., 2017) and EGNN (Satorras et al., 2021) as well as state-of-the-art models for 3D dynamics prediction including EGNO (Xu et al., 2024), HEGNN (Cen et al., 2024) and GF-NODE (Sun et al., 2025). For S2S task, we compare with all of these models; for

Table 1: Test MSE ($\downarrow$), training GPU memory cost and training time per epoch on Motion Capture. F-MSE is reported for `S2S` ($T = 1$) and A-MSE for `S2T` ($T = 5$). We mark the test performance of ⬚ our model and emphasize the best and the runner-up in **bold** and underlined. All improvements with significance level $p < 0.05$ by Wilcoxon signed-rank test are marked with $*$.

| Model | **Walk** (`S2S`) | | | **Run** (`S2S`) | | |
|---|---|---|---|---|---|---|
| | F-MSE ($\times 10^{-2}$) | GPU (GB) | Time (s) | F-MSE ($\times 10^{-1}$) | GPU (GB) | Time (s) |
| Linear (Satorras et al., 2021) | $971 \pm 0.01$ | 0.01 | 0.02 | $625 \pm 0.00$ | 0.01 | 0.03 |
| RF (Köhler et al., 2019) | $93.1 \pm 5.28$ | 0.06 | 0.12 | $9.06 \pm 0.21$ | 0.06 | 0.11 |
| MPNN (Gilmer et al., 2017) | $27.2 \pm 2.15$ | 0.12 | 0.09 | $7.42 \pm 0.90$ | 0.12 | 0.10 |
| EGNN (Satorras et al., 2021) | $26.0 \pm 4.32$ | 0.21 | 0.19 | $4.21 \pm 0.78$ | 0.21 | 0.22 |
| ClofNet (Du et al., 2022) | $\underline{12.6 \pm 1.21}$ | 0.29 | 0.18 | $8.42 \pm 0.69$ | 0.26 | 0.24 |
| EGNO (Xu et al., 2024) | $14.2 \pm 2.62$ | 0.23 | 0.33 | $4.15 \pm 0.42$ | 0.23 | 0.30 |
| HEGNN (Cen et al., 2024) | $16.8 \pm 5.55$ | 0.25 | 0.42 | $5.48 \pm 0.87$ | 0.25 | 0.39 |
| GF-NODE (Sun et al., 2025) | $15.7 \pm 1.34$ | 0.29 | 0.62 | $\underline{3.87 \pm 0.43}$ | 0.28 | 0.67 |
| **PAINET** | $\mathbf{8.45 \pm 0.23}$ | 0.26 | 0.20 | $\mathbf{3.50 \pm 0.27}$ | 0.24 | 0.21 |

| Model | **Walk** (`S2T`) | | | **Run** (`S2T`) | | |
|---|---|---|---|---|---|---|
| | A-MSE ($\times 10^{-1}$) | GPU (GB) | Time (s) | A-MSE ($\times 10^{-1}$) | GPU (GB) | Time (s) |
| EGNO (Xu et al., 2024) | $1.48 \pm 0.55$ | 0.58 | 0.18 | $\underline{5.70 \pm 1.43}$ | 0.58 | 0.19 |
| HEGNN (Cen et al., 2024) | $2.47 \pm 0.89$ | 0.68 | 0.20 | $9.49 \pm 0.02$ | 0.68 | 0.22 |
| GF-NODE (Sun et al., 2025) | $\underline{1.25 \pm 0.17}$ | 0.77 | 1.10 | $7.49 \pm 1.69$ | 0.77 | 0.76 |
| **PAINET** | $\mathbf{0.86 \pm 0.04}$ | 1.04 | 0.28 | $\mathbf{3.33 \pm 0.12}$ | 1.09 | 0.31 |

`S2T`, since the classic models are not originally designed for this task, we only compare with the latest models EGNO, HEGNN and GF-NODE. For fair comparison, we use the same hyperparameter searching space on validation data for all models (see Appendix F.6 for details).

### 4.1 MOTION CAPTURE

**Dataset and Implementation.** CMU Motion Capture dataset (CMU Graphics Lab, 2003) contains 3D joint trajectories collected from various human motion sequences. Following prior works (Huang et al., 2022; Xu et al., 2024), we uniformly discretize the trajectories and focus on two actions: Subject #35 (*Walk*) and Subject #9 (*Run*). More implementation details are provided in Appendix F.2.

**Results.** The results are reported in Table 1. PAINET consistently achieves the lowest prediction error in all the tasks. For `S2S`, PAINET yields $40.5\%$ and $9.6\%$ relative improvements over the most competitive baseline in *Walk* and *Run*, respectively. For `S2T`, PAINET exceeds all the competitors with $31.2\%$ and $41.5\%$ decrease on A-MSE in the task *Walk* and *Run*, respectively, demonstrating its superior capability for predicting 3D dynamics. Additionally, the model consumes comparable time and GPU memory cost compared to the competitors, which corroborates the advantage of PAINET using the same level of computational costs to achieve superior performance.

### 4.2 MOLECULAR DYNAMICS ON SMALL MOLECULES

**Dataset and Implementation.** MD17 (Chmiela et al., 2017) is a widely used benchmark for simulating molecular trajectories governed by quantum-mechanical forces. Following the common protocol, we evaluate the models with `S2S` ($T = 1$) and `S2T` ($T = 8$), using 500/2000/2000 trajectories for training/validation/test. More implementation details are deferred to Appendix F.3.

**Results.** The results are presented in Table 2 and 3 for `S2S` and `S2T`, respectively. Our model consistently outperforms the competitors across all cases with substantial improvements in quite a few cases. Specifically, the performance improvements (which are all statistically significant with at least $p < 0.05$) range from $1.6\%$ to $18.0\%$ and notably, the improvements on *benzene* for `S2S` and *naphthalene* for `S2T` are $13.4\%$ and $18.0\%$, respectively. As demonstrated in Fig. 2, the predicted 3D trajectories of PAINET preserve the essential structural characteristics and contribute to steady results in long time horizons in contrast with the competitor GF-NODE.

### 4.3 MOLECULAR DYNAMICS ON LARGE MOLECULES

**Dataset and implementation.** While MD17 contains mainly small organic molecules, we conducted experiments on MD22 (Chmiela et al., 2023) for evaluating the modeling of complex non-local interactions in larger systems adequately. We evaluate the models with `S2S` ($T = 1$) and `S2T` ($T = 5$), using 500/2000/2000 trajectories for training/validation/test. More implementation details are deferred to Appendix F.3.

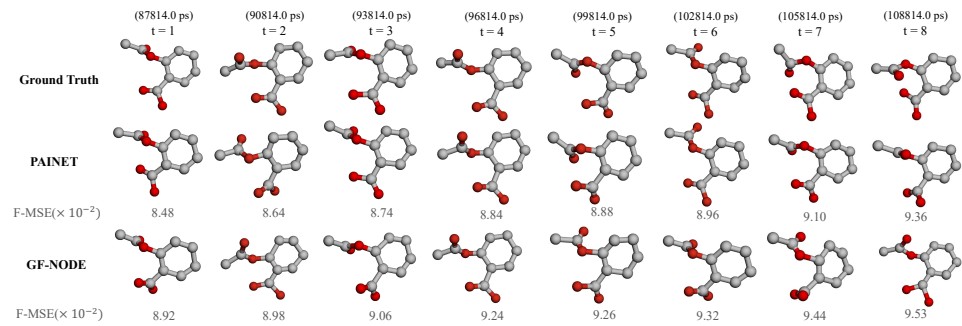

Figure 2: Representative snapshots of aspirin molecular dynamics: the top row shows the ground-truth trajectories, the middle row shows the predictions from PAINET and the bottom row shows the predictions from GF-NODE, with corresponding F-MSEs reported across time steps (where in the parenthesis we show the actual time stamps). More results are presented in Appendix F.10.

Table 2: Test F-MSE ($\downarrow$) for S2S ($T = 1$) on MD17.

| Model | S2S ($T = 1$) | | | |
|---|---|---|---|---|
| | **aspirin** ($\times 10^{-2}$) | **benzene** ($\times 10^{-1}$) | **ethanol** ($\times 10^{-2}$) | **malonaldehyde** ($\times 10^{-1}$) |
| Linear (Satorras et al., 2021) | $12.0 \pm 0.000$ | $16.7 \pm 0.000$ | $5.62 \pm 0.000$ | $2.14 \pm 0.000$ |
| MPNN (Gilmer et al., 2017) | $9.63 \pm 0.053$ | $5.73 \pm 0.569$ | $4.92 \pm 0.084$ | $1.31 \pm 0.004$ |
| RF (Köhler et al., 2019) | $10.9 \pm 0.003$ | $12.6 \pm 0.028$ | $4.64 \pm 0.001$ | $1.29 \pm 0.006$ |
| EGNN (Satorras et al., 2021) | $10.2 \pm 0.158$ | $6.41 \pm 0.246$ | $4.65 \pm 0.005$ | $1.28 \pm 0.001$ |
| ClofNet (Du et al., 2022) | $9.46 \pm 0.029$ | $4.81 \pm 0.010$ | $4.62 \pm 0.006$ | $1.28 \pm 0.001$ |
| EGNO (Xu et al., 2024) | $9.44 \pm 0.122$ | $6.20 \pm 0.248$ | $4.63 \pm 0.004$ | $1.28 \pm 0.001$ |
| HEGNN (Cen et al., 2024) | $9.38 \pm 0.056$ | $5.37 \pm 0.620$ | $4.64 \pm 0.001$ | $1.29 \pm 0.002$ |
| GF-NODE (Sun et al., 2025) | $9.15 \pm 0.030$ | $6.23 \pm 0.602$ | $4.63 \pm 0.001$ | $1.28 \pm 0.000$ |
| **PAINET** | $\mathbf{8.98 \pm 0.040}$ | $\mathbf{4.65 \pm 0.022}$ | $\mathbf{4.31 \pm 0.004}$ | $\mathbf{1.26 \pm 0.001}$ |

| Model | S2S ($T = 1$) | | | |
|---|---|---|---|---|
| | **naphthalene** ($\times 10^{-3}$) | **salicylic** ($\times 10^{-3}$) | **toluene** ($\times 10^{-1}$) | **uracil** ($\times 10^{-3}$) |
| Linear (Satorras et al., 2021) | $6.26 \pm 0.000$ | $13.8 \pm 0.000$ | $1.24 \pm 0.000$ | $9.68 \pm 0.000$ |
| MPNN (Gilmer et al., 2017) | $4.67 \pm 0.036$ | $9.41 \pm 0.023$ | $1.06 \pm 0.042$ | $6.35 \pm 0.028$ |
| RF (Köhler et al., 2019) | $3.98 \pm 0.029$ | $12.4 \pm 0.105$ | $1.09 \pm 0.001$ | $6.21 \pm 0.050$ |
| EGNN (Satorras et al., 2021) | $3.92 \pm 0.042$ | $10.4 \pm 0.154$ | $1.07 \pm 0.015$ | $5.84 \pm 0.082$ |
| ClofNet (Du et al., 2022) | $3.90 \pm 0.066$ | $10.3 \pm 0.581$ | $1.03 \pm 0.002$ | $6.21 \pm 0.415$ |
| EGNO (Xu et al., 2024) | $3.80 \pm 0.077$ | $8.99 \pm 0.641$ | $1.06 \pm 0.005$ | $5.52 \pm 0.149$ |
| HEGNN (Cen et al., 2024) | $3.72 \pm 0.037$ | $8.37 \pm 0.021$ | $1.05 \pm 0.004$ | $5.30 \pm 0.047$ |
| GF-NODE (Sun et al., 2025) | $3.73 \pm 0.166$ | $8.61 \pm 0.254$ | $1.04 \pm 0.002$ | $5.77 \pm 0.017$ |
| **PAINET** | $\mathbf{3.36 \pm 0.008}$ | $\mathbf{7.80 \pm 0.007}$ | $\mathbf{1.00 \pm 0.000}$ | $\mathbf{5.14 \pm 0.004}$ |

**Results.** The results are reported in Table 4 for S2S and S2T, respectively. Across the two molecular systems, PAINET achieves the best performance under all settings, outperforming strong baselines. These improvements demonstrate the effectiveness of our all-pair interaction modeling in larger and structurally diverse molecules.

### 4.4 PROTEINS DYNAMICS

**Dataset and implementation.** The Adenylate Kinase (Adk) Equilibrium dataset (Seyler & Beckstein, 2017) is a long-time-scale protein dynamics simulation of *apo Adk* as a common benchmark for testing long-range structured modeling and scalability to large graphs. We randomly split the dataset into training/validation/testing with the ratio of 3:1:1. To assess the generalization capability of the model, we train all the models with one-step prediction ($T = 1$) and test them with multi-step prediction ($T = 5$). More details are illustrated in Appendix F.5.

**Results.** The results are reported in Table 5 with test MSE of each time step and the overall A-MSE. Compared to all competitors, PAINET achieves the best performance across all intermediate prediction steps (from $t = 1$ to $t = 5$). We also find that the performance improvement by PAINET enlarges as time step increases, which shows that PAINET possesses better capability for modeling the dynamics mechanism that generalizes to simulations in long time horizons.

### 4.5 ABLATION STUDIES

We next conduct a series of ablation studies to analyze the effectiveness of the main components, the impact of key hyperparameters as well as evaluations on the model's scalability.

Table 3: Test A-MSE ($\downarrow$) for S2T ($T = 8$) on MD17.

| Model | S2T ($T = 8$) | | | |
|---|---|---|---|---|
| | **aspirin** ($\times 10^{-2}$) | **benzene** ($\times 10^{-1}$) | **ethanol** ($\times 10^{-2}$) | **malonaldehyde** ($\times 10^{-1}$) |
| EGNO (Xu et al., 2024) | $9.34 \pm 0.049$ | $5.87 \pm 0.182$ | $\underline{4.63 \pm 0.002}$ | $1.28 \pm 0.001$ |
| HEGNN (Cen et al., 2024) | $9.54 \pm 0.024$ | $7.02 \pm 0.233$ | $4.67 \pm 0.001$ | $1.29 \pm 0.001$ |
| GF-NODE (Sun et al., 2025) | $\underline{9.25 \pm 0.017}$ | $\underline{5.48 \pm 0.228}$ | $4.63 \pm 0.006$ | $\underline{1.28 \pm 0.000}$ |
| **PAINET** | $\mathbf{8.84 \pm 0.045}$ | $\mathbf{4.87 \pm 0.295}$ | $\mathbf{4.55 \pm 0.001}$ | $\mathbf{1.26 \pm 0.001}$ |

| Model | S2T ($T = 8$) | | | |
|---|---|---|---|---|
| | **naphthalene** ($\times 10^{-3}$) | **salicylic** ($\times 10^{-3}$) | **toluene** ($\times 10^{-1}$) | **uracil** ($\times 10^{-3}$) |
| EGNO (Xu et al., 2024) | $\underline{3.95 \pm 0.055}$ | $\underline{8.51 \pm 0.070}$ | $1.06 \pm 0.009$ | $\underline{5.68 \pm 0.167}$ |
| HEGNN (Cen et al., 2024) | $4.27 \pm 0.001$ | $8.99 \pm 0.024$ | $1.06 \pm 0.007$ | $6.08 \pm 0.062$ |
| GF-NODE (Sun et al., 2025) | $4.58 \pm 0.065$ | $8.61 \pm 0.009$ | $\underline{1.03 \pm 0.002}$ | $5.82 \pm 0.008$ |
| **PAINET** | $\mathbf{3.24 \pm 0.119}$ | $\mathbf{7.88 \pm 0.142}$ | $\mathbf{1.00 \pm 0.002}$ | $\mathbf{5.06 \pm 0.011}$ |

Table 4: Test MSE ($\downarrow$) on MD22. F-MSE is reported for S2S ($T = 1$) and A-MSE for S2T ($T = 5$). We mark the test performance of our model and emphasize the best and the runner-up in **bold** and underlined.

| Model | *stachyose* | | *Ac-Ala3-NHMe* | |
|---|---|---|---|---|
| | F-MSE ($\times 10^{-1}$)(S2S) | A-MSE ($\times 10^{-1}$)(S2T) | F-MSE ($\times 10^{-1}$)(S2S) | A-MSE ($\times 10^{-1}$)(S2T) |
| EGNN (Satorras et al., 2021) | $3.40 \pm 0.07$ | - | $9.62 \pm 0.14$ | - |
| EGNO (Xu et al., 2024) | $3.33 \pm 0.05$ | $4.43 \pm 0.11$ | $9.11 \pm 0.17$ | $8.92 \pm 0.12$ |
| HEGNN (Cen et al., 2024) | $3.34 \pm 0.14$ | $3.81 \pm 0.20$ | $12.41 \pm 0.53$ | $11.63 \pm 1.07$ |
| GF-NODE (Sun et al., 2025) | $\underline{3.11 \pm 0.27}$ | $\underline{2.54 \pm 0.17}$ | $\underline{9.03 \pm 0.31}$ | $\underline{8.52 \pm 0.07}$ |
| **PAINET** | $\mathbf{3.00 \pm 0.06}$ | $\mathbf{2.40 \pm 0.11}$ | $\mathbf{8.83 \pm 0.18}$ | $\mathbf{8.35 \pm 0.27}$ |

Table 5: Test MSE ($\downarrow$), inference GPU memory cost and inference time cost under multi-step prediction on Adk protein dynamics. All models are trained for single-step prediction and tested for five-step prediction. We report the MSE at time step $t = 1, ..., 5$ and the overall A-MSE.

| Model | S2T ($T = 5$) | | | | | | | |
|---|---|---|---|---|---|---|---|---|
| | **MSE**($t = 1$) | **MSE**($t = 2$) | **MSE**($t = 3$) | **MSE**($t = 4$) | **MSE**($t = 5$) | **A-MSE** | **GPU (GB)** | **Time (s)** |
| EGNO (Xu et al., 2024) | 1.119 | 1.620 | 1.905 | 2.139 | 2.240 | 1.805 | 5.87 | 14.22 |
| HEGNN (Cen et al., 2024) | 1.104 | $\underline{1.611}$ | $\underline{1.875}$ | $\underline{1.998}$ | $\underline{2.087}$ | $\underline{1.735}$ | 8.24 | 18.22 |
| GF-NODE (Sun et al., 2025) | $\underline{1.095}$ | 1.642 | 1.913 | 2.047 | 2.113 | 1.762 | 9.91 | 27.71 |
| **PAINET** | $\mathbf{1.076}$ | $\mathbf{1.608}$ | $\mathbf{1.753}$ | $\mathbf{1.840}$ | $\mathbf{1.994}$ | $\mathbf{1.654}$ | 11.10 | 13.59 |

**Learnable pairwise mappings outperform fixed counterparts.** We compare with multiple variants of PAINET in Fig. 3a where we replace learnable $\mathbf{\Psi}$ and $\mathbf{\Phi}$ with fixed counterparts (e.g., $\mathbf{\Psi} = \mathbf{1}$ using an all-one matrix). The results show that using learnable pairwise mappings yields the optimal A-MSE. In Fig. 3a, we also compare with a simplified version of PAINET that entirely removes the attention module. This simplified model leads to clear performance degradation. These results altogether validate the necessity and superiority of our proposed attention network.

**Parallel equivariant decoder balances accuracy and efficiency.** To assess the efficacy of the equivariant decoder, we substitute it (i.e., the decoder module in Eqn. 2) with other alternatives and compare with PAINET in Fig. 3b. 1) *MLP-add*: we add $\mathbf{X}^{(0)}$ and $\mathbf{H}^{(t)}$ and then use a MLP to predict $\widehat{\mathbf{X}}^{(t)}$. 2) *MLP-concat*: we concatenate $\mathbf{X}^{(0)}$ and $\mathbf{H}^{(t)}$ and then use a MLP to predict $\widehat{\mathbf{X}}^{(t)}$. 3) *EGNN-recurrent*: we change EGNN in our model to a recurrent counterpart that generates $\widehat{\mathbf{X}}^{(t)}$ iteratively from $t = 1$ to $t = T$. The results show that while using MLP is more efficient than EGNN, it suffers from degraded performance. This is intuitive since MLP fails to accommodate the observed structural information. Also, compared with *EGNN-recurrent*, our model yields lower A-MSE with much fewer time costs, which shows the advantage of the parallel decoder in terms of both effectiveness and efficiency.

**Moderate decoder depth balances accuracy and efficiency.** We vary the number of decoding layers (i.e., the number of EGNN layers) from 1 to 10 while keeping all the other factors the same and report the results in Fig. 5. We find that as the number of decoder layers increases, the test F-MSE first decreases and then remains a constant level, while the training time cost linearly increases. This suggests that using a moderate number of decoder layers (e.g., 3) yields the optimal prediction results and in the meanwhile brings up desired efficiency.

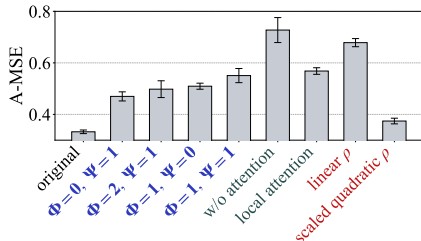 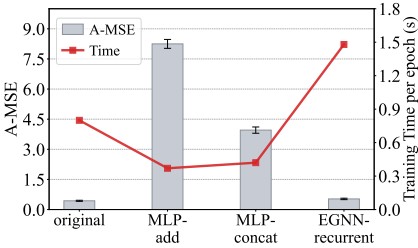

(a) Comparison with attention variants.  (b) Comparison with decoder variants.

Figure 3: Ablation studies w.r.t. learnable pairwise mappings in the attention network and the parallel equivariant decoder on Motion Capture *Run*. In (a), we group the model variants for ablation studies into three categories: 1) ablation on $\phi$ and $\psi$ (in blue), 2) ablation on all-pair attention (in green), and 3) ablation on potential function $\rho$ (in red).

Table 6: Test MSE ($\downarrow$) for different time steps on Motion Capture. F-MSE is reported for $T = 1$ and A-MSE for $T = 5, 10, 15, 20$.

| Model | Walk ($\times 10-1$) | | | | |
|---|---|---|---|---|---|
| | T=1 | T=5 | T=10 | T=15 | T=20 |
| EGNO (Xu et al., 2024) | $1.24 \pm 0.12$ | $1.28 \pm 0.06$ | $1.70 \pm 0.09$ | $7.07 \pm 0.32$ | $7.19 \pm 0.22$ |
| HEGNN (Cen et al., 2024) | $0.83 \pm 0.04$ | $1.42 \pm 0.13$ | $4.15 \pm 0.21$ | $8.41 \pm 0.26$ | $14.3 \pm 0.47$ |
| GF-NODE (Sun et al., 2025) | $1.49 \pm 0.10$ | $1.62 \pm 0.08$ | $1.47 \pm 0.09$ | $2.22 \pm 0.26$ | $1.32 \pm 0.17$ |
| **PAINET** | $\mathbf{0.85 \pm 0.02}$ | $\mathbf{0.86 \pm 0.03}$ | $\mathbf{1.20 \pm 0.05}$ | $\mathbf{1.33 \pm 0.07}$ | $\mathbf{1.36 \pm 0.04}$ |

| Model | Run ($\times 10-1$) | | | | |
|---|---|---|---|---|---|
| | T=1 | T=5 | T=10 | T=15 | T=20 |
| EGNO (Xu et al., 2024) | $4.16 \pm 0.33$ | $5.89 \pm 0.52$ | $5.31 \pm 0.41$ | $5.88 \pm 0.48$ | $6.79 \pm 0.62$ |
| HEGNN (Cen et al., 2024) | $5.33 \pm 0.74$ | $9.49 \pm 0.89$ | $671 \pm 6.92$ | $118 \pm 9.21$ | $681 \pm 9.72$ |
| GF-NODE (Sun et al., 2025) | $3.72 \pm 0.17$ | $7.49 \pm 0.22$ | $4.44 \pm 0.30$ | $6.34 \pm 0.41$ | $6.62 \pm 0.21$ |
| **PAINET** | $\mathbf{3.50 \pm 0.27}$ | $\mathbf{3.33 \pm 0.12}$ | $\mathbf{4.34 \pm 0.18}$ | $\mathbf{5.42 \pm 0.27}$ | $\mathbf{5.96 \pm 0.17}$ |

**One attention layer per time step suffices.** We next study the impact of attention layers per time step and change its number from 1 to 6 in Fig. 6. We find that the optimal performance is achieved by using one attention layer per time step, in which case the time cost is minimal in contrast with the model using multiple attention layers per time step. The indicates that one-layer attention is expressive enough for capturing all-pair interactions among arbitrary particles and compared to the multi-layer counterpart, this simpler model is easier for optimization and less prone to over-fitting.

**Robustness persists across time steps.** To assess robustness, we further evaluate prediction across various time steps. As shown in Table 6, PAINET consistently performs best on the Motion Capture dataset, demonstrating strong stability in long trajectory prediction.

**Computation costs scale nearly linearly w.r.t. particle numbers and time steps.** To test the scalability of PAINET, we vary the number of time steps and particles on Proteins (Adk) and report the inference GPU memory cost and time cost in Fig. 7. We find that the GPU memory usage exhibits a linear increasing trend w.r.t. time steps for prediction as well as particle numbers. Moreover, Table 9 compares training and inference time across models, where PAINET achieves competitive efficiency. This demonstrates the desired scalability of PAINET to large-scale multi-body systems.

## 5 CONCLUSION

We have proposed PAINET, a physics-inspired equivariant architecture, for modeling 3D dynamics. PAINET consists of a principled attention layer grounded on the principle of energy minimization and a parallel equivariant decoder to maintain SE(3)-equivariance and inference efficiency. This architecture addresses the key limitation in existing graph-based dynamics models, which often rely solely on local observed structures. Various experiments across diverse domains, including motion capture, molecular dynamics, and protein dynamics, demonstrate that our model consistently outperforms prior models across various benchmarks with desired efficiency.

ACKNOWLEDGEMENT

The research was supported by Shanghai Artificial Intelligence Laboratory. Additionally, We thank the open-source community for releasing code and datasets that made this research possible.

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

## A    RELATED WORKS

We briefly discuss the relevant literature to include more background.

**3D Dynamics Prediction.** Predicting 3D dynamics—such as particle motion, robotic trajectories, or molecular interactions—is a fundational challenge in physics simulation and robotics. Traditional physics-based models offer interpretability but often fall short when modeling complex or unknown interactions. More recently, data-driven approaches, especially deep learning, have shown strong potential (Karniadakis et al., 2021; Wu et al., 2024; Huang et al., 2024). While RNNs, LSTMs, and Transformers can model temporal dependencies, they often fail to capture the underlying relational structure in multi-agent or multi-body systems (Kipf et al., 2018). Graph Neural Networks (GNNs) address this by modeling entities as nodes and their interactions as edges (Battaglia et al., 2016; Han et al., 2025). Recent advances in Equivariant GNNs (EGNNs) (Satorras et al., 2021; Huang et al., 2022; Han et al., 2022) further improve physical fidelity by preserving Euclidean symmetries, enabling more accurate and generalizable 3D dynamics predictions. Some subsequent works focus on the efficiency of EGNNs through architectural optimizations (Zhang et al., 2024).

**Message Passing Neural Networks.** Graph neural networks (GNNs) (Gilmer et al., 2017; Kipf & Welling, 2017) provide a general framework for learning over relational structures and have recognized as a powerful method for simulating physical systems. One line of related works harness physical priors to design more expressive message-passing operators to encode system interactions (Mrowca et al., 2018; Shi et al., 2025; Viswanath et al., 2024) or incorporate classical physical mechanics into the architecture (Sanchez-Gonzalez et al., 2019). A parallel line of works focus on encoding the symmetries of Euclidean space, i.e., equivariance, as an inductive bias into the architectures, including translation equivariance (Ummenhofer et al., 2019; Sanchez-Gonzalez et al., 2020; Pfaff et al., 2021), full rotation and reflection equivariance through spherical harmonics (Thomas et al., 2018; Fuchs et al., 2020) or equivariant message passing (Satorras et al., 2021; Huang et al., 2022; Thiemann et al., 2025). Further refinements exploit local coordinate frames to process higher-order geometric features (Liu et al., 2022; Du et al., 2022; Han et al., 2024; Cen et al., 2024) or model the temporal information by Fourier transform (Xu et al., 2024) while preserving equivariance. Recently, a neural-ODE (Sun et al., 2025) formulation to enhance expressivity. Despite these advances, these EGNNs are constrained on the observed structure and neglect to incorporate the latent all-pair interactions, which is crucial in physics.

## B    EQUIVARIANCE

In this work, we consider equivariance properties with respect to the special Euclidean group $\text{SE}(3)$, which includes translations, rotations and permutations. For 3D dynamics, the three types of equivariance considered are summarized below:

Table 7: Three types of equivariance and corresponding conditions.

| Type | Transformation | Condition |
|------|---------------|-----------|
| Translation | $\mathbf{g} \in \mathbb{R}^3$ | $\mu(\mathbf{X} + \mathbf{g}) = \mu(\mathbf{X}) + \mathbf{g}$ |
| Rotation | $\mathbf{Q} \in \mathbb{R}^{3\times3}$ | $\mu(\mathbf{QX}) = \mathbf{Q}\mu(\mathbf{X})$ |
| Permutation | $\mathbf{P} \in \mathbb{R}^{n\times n}$ | $\mu(\mathbf{PX}) = \mathbf{P}\mu(\mathbf{X})$ |

Here $\mathbf{X} = [\mathbf{x}_1^T; \ldots; \mathbf{x}_N^T] \in \mathbb{R}^{N\times3}$ denotes a set of position vectors for $N$ nodes in 3D space, and $\mu : \mathbb{R}^{N\times3} \to \mathbb{R}^{N\times3}$ is a function on $\mathbf{X}$. As illustrated in Fig. 4, translation equivariance implies that a uniform shift of all input positions by a vector $\mathbf{g}$ results in the output being shifted by the same amount. Rotation equivariance indicates that applying an orthogonal transformation $\mathbf{Q}$ to the input causes the output to be rotated in the same way. Permutation equivariance means that reordering the input points leads to an identically permuted output.

Note that for velocity vectors $\mathbf{V} = [\mathbf{v}_1^T; \ldots; \mathbf{v}_N^T] \in \mathbb{R}^{N\times3}$ and $\xi : \mathbb{R}^{N\times3} \to \mathbb{R}^{N\times3}$, translations do not alter their values. However, they transform equivariantly under rotations and permutations, i.e.,

$$\mathbf{Q}\xi(\mathbf{V}) = \xi(\mathbf{QV}), \quad \mathbf{P}\xi(\mathbf{V}) = \xi(\mathbf{PV}).$$

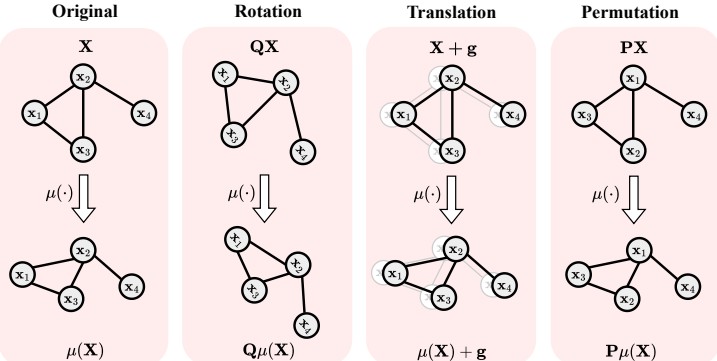

Figure 4: Illustration for three types of equivariance, including rotation, translation and permutation.

These equivariance properties are essential in modeling systems that comply with spatial symmetries, such as molecular dynamics or physical simulations.

## C   ENERGY MINIMIZATION ON GRAPHS

Graph representation learning can be rigorously formulated as an optimization problem governed by an energy minimization framework, where node embeddings evolve to satisfy smoothness and consistency criteria. Modern Graph Neural Networks (GNNs) can be interpreted as iterative solvers for this energy function. Foundational approaches establish this through the minimization of quadratic energy functionals to enforce local and global consistency on the graph manifold. It has been shown that the propagation rules in standard architectures, such as GCN (Jiang et al., 2019) and SGC (Wu et al., 2019), effectively perform gradient descent steps on an objective energy functionals with a fitting term (Ma et al., 2021). Extending this theoretical framework to modeling 3D dynamics with latent structures, we characterize the uncovering of unobserved interactions as the minimization of a regularized energy function. Formally, this energy $E$ at layer $t$ balances feature fidelity with structural smoothness:

$$E(H, t; \{\rho_{ij}\}) = \sum_i ||h_i - h_i^{(t)}||_2^2 + \lambda \sum_{i,j} \rho_{ij}(||h_i - h_j||_2^2), \tag{12}$$

where $H$ represents the stack of particle embeddings, $\rho_{ij}$ denotes a non-linear potential function, and $\lambda$ is a regularization weight. The first term serves as a fidelity constraint, regularizing the deviation of updated embeddings from their previous states, while the second term penalizes pairwise distances to enforce internal consistency among particles. This formulation draws inspiration from physical systems where state evolution naturally transitions from high-energy to low-energy regimes to reach equilibrium. By instantiating $\rho_{ij}$ as a Landau-Ginzburg potential, this framework connects the optimization process to phenomenological explanations of matter formation and symmetry breaking. Consequently, the minimization of this energy yields a principled attention mechanism where propagation steps correspond to rigorous gradient descent updates.

## D   PROOF FOR THEOREM 1

We first convert the minimization of Eqn. 3 into the minimization of its variational upper bound. For any concave, non-decreasing function $\rho : \mathbb{R}^+ \to \mathbb{R}$, one can express it as the variational decomposition

$$\rho(z^2) = \min_{\omega \geq 0}[\omega z^2 - \tilde{\rho}(\omega)] \geq \omega z^2 - \tilde{\rho}(\omega), \tag{13}$$

where $\omega$ is a variational parameter and $\tilde{\rho}$ is the concave conjugate of $\rho$. Eqn. 13 defines $\rho(z^2)$ as the minimal envelope of a series of quadratic bounds $\omega z^2 - \tilde{\rho}(\omega)$ defined by a different values of $\omega \geq 0$. The upper bound is given for a fixed $\omega$ when removing the minimization operator. We note that for any optimal $\omega^*$ we have

$$\omega^* z^2 - \tilde{\rho}(\omega^*) = \rho(z^2), \tag{14}$$

which is tangent to $\rho$ at $z^2$ and $\omega^* = \frac{\partial \delta(z^2)}{\partial z^2}$. This gives the sufficient and necessary condition for the equality of Eqn. 13.

Based on the above result, we can derive the variational upper bound of Eqn. 3:

$$\tilde{E}(\mathbf{H}, t; \{\omega_{ij}\}, \{\tilde{\rho}_{ij}\}) = \sum_i \|\mathbf{h}_i - \mathbf{h}_i^{(t)}\|_2^2 + \lambda \left[ \sum_{i,j} \omega_{ij} \|\mathbf{h}_i - \mathbf{h}_j\|_2^2 - \tilde{\rho}(\omega_{ij}) \right], \qquad (15)$$

where $\tilde{\rho}$ is the concave conjugate of $\rho$, and the equality holds if and only if the variational parameters satisfy

$$\omega_{ij} = \left. \frac{\partial \rho_{ij}(h^2)}{\partial h^2} \right|_{h = \|\mathbf{h}_i - \mathbf{h}_j\|_2}. \qquad (16)$$

In light of this relationship, we can minimize the upper bound surrogate Eqn. 15 which is equivalent to the minimization of Eqn. 3 on condition that the variational parameters $\omega_{ij}$'s are given by Eqn. 16. Then, consider a gradient decent step on Eqn. 15, the particle embeddings are updated through (assuming $\frac{\tau}{2}$ as the step size)

$$\begin{aligned}
\mathbf{H}^{(t+1)} &= \mathbf{H}^{(t)} - \tau \left. \frac{\partial \tilde{E}(\mathbf{H}, t; \{\omega_{ij}\}, \{\tilde{\rho}_{ij}\})}{\partial \mathbf{H}} \right|_{\mathbf{H} = \mathbf{H}^{(t)}} \\
&= \mathbf{H}^{(t)} - \tau \left( \lambda (\mathbf{D}^{(t)} - \mathbf{\Omega}^{(t)}) \mathbf{H}^{(t)} + \mathbf{H}^{(t)} - \mathbf{H}^{(t)} \right) \\
&= \mathbf{H}^{(t)} - \tau \lambda (\mathbf{D}^{(t)} - \mathbf{\Omega}^{(t)}) \mathbf{H}^{(t)},
\end{aligned} \qquad (17)$$

where $\mathbf{\Omega}^{(t)} = \{\omega_{ij}^{(t)}\}_{N \times N}$ and $\mathbf{D}^{(t)}$ denotes the diagonal degree matrix associated with $\mathbf{\Omega}^{(t)}$. Common practice to accelerate convergence adopts a positive definite preconditioner term, e.g., $(\mathbf{D}^{(t)})^{-1}$, to re-scale the updating gradient and the final updating form becomes

$$\mathbf{H}^{(t+1)} = (1 - \tau \lambda) \mathbf{H}^{(t)} + \tau \lambda (\mathbf{D}^{(t)})^{-1} \mathbf{\Omega}^{(l)} \mathbf{H}^{(l)}. \qquad (18)$$

The above iteration converges for step size $0 < \tau < \frac{1}{\lambda}$. We thus prove that

$$E(\mathbf{H}^{(t)}, t; \{\rho_{ij}\}) \geq E(\mathbf{H}^{(t+1)}, t; \{\rho_{ij}\}). \qquad (19)$$

Besides, we notice that for a fixed $\mathbf{H}$, $E(\mathbf{H}, t; \{\rho_{ij}\}) = \|\mathbf{H} - \mathbf{H}^{(t)}\|_F^2 + \lambda \sum_{ij} \rho_{ij}(\|\mathbf{h}_i - \mathbf{h}_j\|_2^2)$ becomes a function of $t$ and its optimum is achieved if and only if $\mathbf{H}^{(t)} = \mathbf{H}$. Such a fact yields that

$$E(\mathbf{H}^{(t)}, t; \{\rho_{ij}\}) \geq E(\mathbf{H}^{(t)}, t+1; \{\rho_{ij}\}). \qquad (20)$$

The result of the main theorem follows by noting that $E(\mathbf{H}^{(t)}, t; \{\rho_{ij}\}) \geq E(\mathbf{H}^{(t+1)}, t; \{\rho_{ij}\}) \geq E(\mathbf{H}^{(t+1)}, t+1; \{\rho_{ij}\})$.

## E    PROOF FOR EQUIVARIANCE OVER PAINET

We provide a formal proof that our model architecture preserves SE(3)-equivariance throughout the trajectory prediction process. Let $\mathbf{Q} \in SO(3)$ be a rotation matrix and $\mathbf{g} \in \mathbb{R}^3$ a translation vector. A function $f(\cdot)$ is said to be *equivariant* to SE(3) transformations if, for all positions $\mathbf{X}$ and all transformations $(\mathbf{Q}, \mathbf{g})$, it holds that:

$$f(\mathbf{Q}\mathbf{X} + \mathbf{g}) = \mathbf{Q} f(\mathbf{X}) + \mathbf{g}. \qquad (21)$$

Similarly, a function is said to be *invariant* if:

$$f(\mathbf{Q}\mathbf{X} + \mathbf{g}) = f(\mathbf{X}). \qquad (22)$$

Our model consists of two major components: (1) a physics-inspired attention network that operates on particle embeddings $\mathbf{h}_i^{(t)}$'s, and (2) an equivariant decoder that updates positions $\mathbf{x}_i^{(t)}$ using messages aggregated by both $\mathbf{h}_i^{(t)}$ and relative positions $\mathbf{x}_i - \mathbf{x}_j$. We can prove that, the attention network outputs SE(3)-**invariant** embeddings $\mathbf{h}_i^{(t)}$, and the decoder is SE(3)-**equivariant** with respect to its positional inputs.

**Invariance of embeddings.** At each time step, the attention network operates merely on latent embeddings $\mathbf{h}_i^{(t)}$'s, which are independent of the input coordinates $\mathbf{x}_i$. The attention mechanism computes dot-product similarity scores (e.g., $\mathbf{h}_i^\top \mathbf{h}_j$), and produces updates with adaptive pairwise mappings. Since $\mathbf{h}_i$'s are defined in latent space and do not transform under SE(3), their updates are invariant. For instance, as for rotation operation, we have:

$$\forall \mathbf{Q} \in SO(3), \quad \mathbf{h}_i^{(t)} = \textit{Attention}(\mathbf{F}; \mathbf{Q}\mathbf{X}^{(0)}) = \textit{Attention}(\mathbf{F}) = \textit{Attention}(\mathbf{F}; \mathbf{X}^{(0)}), \quad (23)$$

where $\mathbf{Q}$ is an orthogonal rotation matrix and $\mathbf{F}$ is the initial node features.

**Equivariance of decoder.** The decoder is composed of equivariant GNN layers. Each EGNN layer updates positions using the form:

$$\begin{aligned}
\mathbf{m}_{ij}^{(l)} &= \phi_m(\mathbf{h}_i^{(l)}, \mathbf{h}_j^{(l)}, \|\mathbf{x}_i^{(l)} - \mathbf{x}_j^{(l)}\|^2, \mathbf{a}_{ij}), \\
\mathbf{h}_i^{(l+1)} &= \phi_h\left(\mathbf{h}_i^{(l)}, \sum_{j \in \mathcal{N}(i)} \mathbf{m}_{ij}^{(l)}\right), \\
\mathbf{x}_i^{(l+1)} &= \mathbf{x}_i^{(l)} + \sum_{j \in \mathcal{N}(i)} (\mathbf{x}_i^{(l)} - \mathbf{x}_j^{(l)}) \cdot \phi_x(\mathbf{m}_{ij}^{(l)}),
\end{aligned} \quad (24)$$

where $\phi_x, \phi_m, \phi_h$ are MLPs. For any orthogonal rotation transformation $\mathbf{Q} \in SO(3)$ and $\mathbf{g} \in \mathbb{R}^3$, we can prove that $\|\mathbf{x}_i^{(l)} - \mathbf{x}_j^{(l)}\|^2$ is invariant, since:

$$\begin{aligned}
\|(\mathbf{Q}\mathbf{x}_i^{(l)} + \mathbf{g}) - (\mathbf{Q}\mathbf{x}_j^{(l)} + \mathbf{g})\|^2 &= \|\mathbf{Q}\mathbf{x}_i^{(l)} - \mathbf{Q}\mathbf{x}_j^{(l)}\|^2 \\
&= (\mathbf{x}_i^{(l)} - \mathbf{x}_j^{(l)})^T \mathbf{Q}^T \mathbf{Q}(\mathbf{x}_i^{(l)} - \mathbf{x}_j^{(l)}) \\
&= \|\mathbf{x}_i^{(l)} - \mathbf{x}_j^{(l)}\|^2.
\end{aligned} \quad (25)$$

Thus, $\mathbf{m}_{ij}$ is invariant. Moreover, we have:

$$\begin{aligned}
\mathbf{Q}\mathbf{x}_i^{(l+1)} + \mathbf{g} &= \mathbf{Q}(\mathbf{x}_i^{(l)} + \sum_{j \in \mathcal{N}(i)} (\mathbf{x}_i^{(l)} - \mathbf{x}_j^{(l)}) \cdot \phi_x(\mathbf{m}_{ij}^{(l)})) + \mathbf{g} \\
&= (\mathbf{Q}\mathbf{x}_i^{(l)} + \mathbf{g}) + \mathbf{Q}\sum_{j \in \mathcal{N}(i)} (\mathbf{x}_i^{(l)} - \mathbf{x}_j^{(l)}) \cdot \phi_x(\mathbf{m}_{ij}^{(l)})) \\
&= (\mathbf{Q}\mathbf{x}_i^{(l)} + \mathbf{g}) + \sum_{j \in \mathcal{N}(i)} (\mathbf{Q}\mathbf{x}_i^{(l)} - \mathbf{Q}\mathbf{x}_j^{(l)}) \cdot \phi_x(\mathbf{m}_{ij}^{(l)})) \\
&= (\mathbf{Q}\mathbf{x}_i^{(l)} + \mathbf{g}) + \sum_{j \in \mathcal{N}(i)} ((\mathbf{Q}\mathbf{x}_i^{(l)} + \mathbf{g}) - (\mathbf{Q}\mathbf{x}_j^{(l)} + \mathbf{g})) \cdot \phi_x(\mathbf{m}_{ij}^{(l)}).
\end{aligned} \quad (26)$$

Since $\mathbf{m}_{ij}$ is invariant and the position updates are equivariant with respect to SE(3) transformations, the overall model preserves equivariance by design:

$$\begin{aligned}
(\widehat{\mathbf{X}}, \widehat{\mathbf{H}}) &= \text{PAINET}(\mathbf{X}^{(0)}; \mathbf{H}^{(0)}), \\
(\mathbf{Q}\widehat{\mathbf{X}} + \mathbf{g}, \widehat{\mathbf{H}}) &= \text{PAINET}(\mathbf{Q}\mathbf{X}^{(0)} + \mathbf{g}; \mathbf{H}^{(0)}).
\end{aligned} \quad (27)$$

# F  EXPERIMENT DETAILS

## F.1  EVALUATION METRICS

All the experiments involve two tasks: *state-to-state* (S2S) and *state-to-trajectory* (S2T). The S2S task directly predicts the final state, while the S2T task predicts the trajectories of a specified number of future time steps. Correspondingly, S2S uses Final Mean Squared Error (F-MSE) to measure the

---

**Algorithm 1** PAINET for 3D dynamics modeling and prediction

---

**Require:** Initial positions $\mathbf{X}^{(0)}$; initial features $\mathbf{F}$; trajectory length $T$; decoder layer number $L$; hyperparameter $\eta$; adaptive pairwise mappings $\boldsymbol{\Phi}, \boldsymbol{\Psi}$.
**Ensure:** Position predictions for each time step $\{\mathbf{X}^{(1)}, \ldots, \mathbf{X}^{(T)}\}$.
1: $\mathbf{H}^{(0)} \leftarrow \mathbf{W}_f \mathbf{F}$
2: **for** $t = 0$ to $T - 1$ **do**
3: $\quad \mathbf{Q}^{(t)}, \mathbf{K}^{(t)}, \mathbf{V}^{(t)} \leftarrow \mathbf{W}_Q^{(t)}\mathbf{H}^{(t)}, \mathbf{W}_K^{(t)}\mathbf{H}^{(t)}, \mathbf{W}_V^{(t)}\mathbf{H}^{(t)}$
4: $\quad$ Compute the L2-normalized query and key matrices $\tilde{\mathbf{Q}}^{(t)}, \tilde{\mathbf{K}}^{(t)}$
5: $\quad \mathbf{D}^{(t+1)} \leftarrow \operatorname{diag}^{-1}\left(\boldsymbol{\Phi}\mathbf{1} + (\boldsymbol{\Psi} \odot (\tilde{\mathbf{Q}}^{(t)}(\tilde{\mathbf{K}}^{(t)})^T))\mathbf{1}\right)$
6: $\quad \mathbf{H}^{(t+1)} \leftarrow (1 - \eta)\mathbf{H}^{(t)} + \eta\mathbf{D}^{(t+1)}\left(\boldsymbol{\Phi}\mathbf{V}^{(t)} + (\boldsymbol{\Psi} \odot (\tilde{\mathbf{Q}}^{(t)}(\tilde{\mathbf{K}}^{(t)})^T))\mathbf{V}^{(t)}\right)$
7: $\quad \mathbf{H}^{(t+1;0)}, \mathbf{X}^{(t+1;0)} \leftarrow \mathbf{H}^{(t+1)}, \mathbf{X}^{(0)}$
8: $\quad$ **for** $l = 0$ to $L - 1$ **do**
9: $\quad\quad \mathbf{X}^{(t+1;l+1)} \leftarrow \text{EGNN}^{(l)}(\mathbf{X}^{(t+1;l)}, \mathbf{H}^{(t+1)})$
10: $\quad$ **end for**
11: $\quad \widehat{\mathbf{X}}^{(t+1)} \leftarrow \mathbf{X}^{(t+1;L)}$
12: **end for**
13: **return** $\{\widehat{\mathbf{X}}^{(1)}, \ldots, \widehat{\mathbf{X}}^{(T)}\}$

---

MSE between the predicted final state and the ground truth, and `S2T` calculates Average MSE (A-MSE) to assess the MSE averaged across all discretized time steps along the decoded trajectory. Specifically, the definitions of two evaluation metrics are:

$$\text{F-MSE} = \frac{1}{N}\sum_{i=1}^{N}\|\hat{\mathbf{x}}_i(T) - \mathbf{x}_i(T)\|^2, \tag{28}$$

$$\text{A-MSE} = \frac{1}{T}\frac{1}{N}\sum_{t=1}^{T}\sum_{i=1}^{N}\|\hat{\mathbf{x}}_i(t) - \mathbf{x}_i(t)\|^2, \tag{29}$$

where $i \in \mathcal{V}, N = |\mathcal{V}|$, $\hat{\mathbf{x}}_i(t)$ denotes the predicted position for particle $i$ at time step $t$, and $\mathbf{x}_i(t)$ denotes the corresponding ground truth.

### F.2 MOTION CAPTURE

We use two subsets from the CMU Motion Capture dataset (CMU Graphics Lab, 2003), which provides high-fidelity 3D trajectories of human joint movements. Specifically, we follow the protocols adopted by Huang et al. (2022); Han et al. (2022), focusing on Subject #35 (*Walk*) and Subject #9 (*Run*). The dataset is partitioned into training, validation, and test sets with 200/600/600 samples for Subject #35 and 200/240/240 for Subject #9, respectively. The trajectories are uniformly discretized by $\Delta T = 30$. Each snapshot in every trajectory consists of 3D coordinates of 31 human joints. Following prior work, joints are modeled as graph nodes, with connections representing physical or kinematic constraints.

The model is conditioned on the initial positions and velocities of joints, and trained to predict future joint positions over a 100-step horizon. Training is performed using the Adam optimizer with a learning rate of $10^{-3}$, minimizing the mean squared error over predicted trajectories. For both training and evaluation, we set $T = 5$ in `S2T` tasks. Namely, for each initial position, the model needs to predict the positions in the next 5 time step.

### F.3 MOLECULAR DYNAMICS ON SMALL MOLECULES (MD17)

We evaluate our model on the MD17 dataset (Chmiela et al., 2017), which provides ab initio molecular dynamics trajectories for several small organic molecules. For each molecule, we randomly partition the dataset into 500 training, 2000 validation, and 2000 test trajectories with designated random seed for reproducibility. Following the common practice, we use a temporal interval of $\Delta T = 3000$ for uniform discretization and dispose of the first and last 10000 snapshots in case of instability. Each training sample consists of a sequence of atomic positions and velocities at the

initial time step, with the task of predicting atomic positions at future time steps. The initial velocity is computed from consecutive positional differences within the trajectory. Following common pre-processing conventions, we focus on heavy atoms by excluding hydrogens from the graph. For the molecular graph structure, we follow prior works (Shi et al., 2021; Xu et al., 2022; 2024) and augment the connectivity by adding 2-hop neighbors. Edge features include a combination of atomic types, bond types, and hop distance between connected atoms.

For both training and evaluation, we set $T = 8$ in S2T tasks. Namely, for each initial position, the model needs to predict the positions in the next 8 time step.

### F.4 Molecular Dynamics on large Molecules (MD22)

MD22 (Chmiela et al., 2023) contains trajectories for significantly larger and more flexible systems (ranging from 42 to 370 atoms), including supramolecules and major biomolecules like Ac-Ala3-NHMe (peptide), DHA (lipid), and Stachyose (carbohydrate), which further validates model effectiveness for predicting the dynamics of larger, more complex molecules. For each molecule, we randomly partition the dataset into 500 training, 2000 validation, and 2000 test trajectories with designated random seed for reproducibility. We use a temporal interval of $\Delta T = 1000$ for uniform discretization and dispose of the first and last 10000 snapshots in case of instability. Each training sample consists of a sequence of atomic positions and velocities at the initial time step, with the task of predicting atomic positions at future time steps. The trajectories were sampled at temperatures between 400 K and 500 K at a resolution of 1 fs, and energies and forces were computed at the PBE+MBD (Perdew et al., 1996) level of theory.

For both training and evaluation, we set $T = 5$ in S2T tasks. Namely, for each initial position, the model needs to predict the positions in the next 5 time step.

### F.5 Protein Dynamics (Adk)

We evaluate our model on the Adk protein equilibrium trajectory dataset, originally introduced by Seyler & Beckstein (2017) and later preprocessed by Han et al. (2022). The simulation captures the molecular dynamics of apo adenylate kinase (Adk) using the CHARMM27 force field (MacKerell Jr et al., 2000), under NPT conditions at 300 K and 1 bar. Water molecules and ions are modeled explicitly, and snapshots are recorded every 240 picoseconds, yielding a total trajectory length of approximately 1.004 microseconds. This dataset is integrated into the MDAnalysis toolkit (Gowers et al., 2019), which facilitates parsing and processing of molecular simulation outputs. We adopt the same data split as MacKerell Jr et al. (2000), with 2481 sub-trajectories for training, 827 for validation, and 878 for testing. The molecular graph is constructed from backbone atoms using a 10 Å cutoff for edge definition. Each sample provides initial atomic positions and velocities as input, and the task is to forecast future atomic configurations.

For generalization, we set the model to train with one-step prediction ($T = 1$), evaluate with multi-step prediction ($T = 5$). Namely, the model is trained to directly predict the final position, but evaluated on prediction of positions in 5 time steps.

### F.6 Hyperparameters

Table 8 summarizes the key hyperparameters used for training PAINET across all datasets, including learning rate (`lr`), weight decay (`weight decay`), batch size (`batch`), number of decoder layers (`layer`), hidden size (`hidden`), and number of attention heads (`num_heads`). We use Adam optimizer and models are trained until convergence with early stopping based on validation loss, using a patience of 50 epochs. All runs use the same random seed to ensure reproducibility.

For Motion Capture and Protein datasets, we use relatively smaller batch sizes due to memory constraints, while MD17 allows larger batches given its lower per-sample dimensionality. For other hyperparameters, we search them on the validation set with the searching space: $\text{lr} \in \{1 \times 10^{-4}, 5 \times 10^{-4}, 1 \times 10^{-3}, 2 \times 10^{-3}\}$, `weight decay` $\in \{1 \times 10^{-10}, 1 \times 10^{-12}, 1 \times 10^{-15}, 1 \times 10^{-18}\}$, `layer` $\in \{3, 4, 5, 6, 7\}$, `hidden` $\in \{32, 64, 128\}$, `num_heads` $\in \{2, 3, 4, 5\}$.

Table 8: Summary of hyperparameter settings of PAINET.

| dataset | time step | `batch` | `lr` | `weight decay` | `layer` | `hidden` | `num_heads` |
|---|---|---|---|---|---|---|---|
| Motion Capture (*Walk*) | 1 | 12 | $1 \times 10^{-4}$ | $1 \times 10^{-15}$ | 7 | 128 | 4 |
| Motion Capture (*Walk*) | 5 | 12 | $5 \times 10^{-4}$ | $1 \times 10^{-15}$ | 7 | 128 | 3 |
| Motion Capture (*Run*) | 1 | 12 | $5 \times 10^{-4}$ | $1 \times 10^{-15}$ | 7 | 128 | 2 |
| Motion Capture (*Run*) | 5 | 12 | $5 \times 10^{-4}$ | $1 \times 10^{-15}$ | 7 | 128 | 4 |
| MD17 (*aspirin*) | 1 | 100 | $5 \times 10^{-4}$ | $1 \times 10^{-15}$ | 7 | 64 | 4 |
| MD17 (*benzene*) | 1 | 100 | $5 \times 10^{-4}$ | $1 \times 10^{-15}$ | 5 | 64 | 2 |
| MD17 (*ethanol*) | 1 | 100 | $5 \times 10^{-4}$ | $1 \times 10^{-15}$ | 7 | 64 | 4 |
| MD17 (*malonaldehyde*) | 1 | 100 | $1 \times 10^{-4}$ | $1 \times 10^{-15}$ | 8 | 64 | 4 |
| MD17 (*naphthalene*) | 1 | 100 | $5 \times 10^{-4}$ | $1 \times 10^{-15}$ | 6 | 64 | 2 |
| MD17 (*salicylic*) | 1 | 100 | $5 \times 10^{-4}$ | $1 \times 10^{-15}$ | 8 | 64 | 3 |
| MD17 (*toluene*) | 1 | 100 | $5 \times 10^{-4}$ | $1 \times 10^{-15}$ | 7 | 64 | 2 |
| MD17 (*uracil*) | 1 | 100 | $1 \times 10^{-4}$ | $1 \times 10^{-15}$ | 7 | 64 | 4 |
| MD17 (*aspirin*) | 8 | 100 | $5 \times 10^{-4}$ | $1 \times 10^{-15}$ | 7 | 64 | 4 |
| MD17 (*benzene*) | 8 | 100 | $5 \times 10^{-4}$ | $1 \times 10^{-15}$ | 5 | 64 | 2 |
| MD17 (*ethanol*) | 8 | 100 | $5 \times 10^{-4}$ | $1 \times 10^{-15}$ | 7 | 64 | 4 |
| MD17 (*malonaldehyde*) | 8 | 100 | $1 \times 10^{-4}$ | $1 \times 10^{-15}$ | 8 | 64 | 4 |
| MD17 (*naphthalene*) | 8 | 100 | $5 \times 10^{-4}$ | $1 \times 10^{-15}$ | 6 | 64 | 2 |
| MD17 (*salicylic*) | 8 | 100 | $5 \times 10^{-4}$ | $1 \times 10^{-15}$ | 8 | 64 | 3 |
| MD17 (*toluene*) | 8 | 100 | $5 \times 10^{-4}$ | $1 \times 10^{-15}$ | 7 | 64 | 2 |
| MD17 (*uracil*) | 8 | 100 | $1 \times 10^{-4}$ | $1 \times 10^{-15}$ | 7 | 64 | 4 |
| Protein | 5 | 4 | $5 \times 10^{-4}$ | $1 \times 10^{-10}$ | 4 | 32 | 2 |

Table 9: Training time per epoch and inference time on Motion Capture.

| Model | *Walk* (S2S) | | *Run* (S2S) | |
|---|---|---|---|---|
| | Training Time (s) | Inference Time (s) | Training Time (s) | Inference Time (s) |
| Linear (Satorras et al., 2021) | 0.02 | 0.07 | 0.03 | 0.03 |
| RF (Köhler et al., 2019) | 0.12 | 0.20 | 0.11 | 0.08 |
| MPNN (Gilmer et al., 2017) | 0.09 | 0.16 | 0.10 | 0.07 |
| EGNN (Satorras et al., 2021) | 0.19 | 0.26 | 0.22 | 0.14 |
| EGNO (Xu et al., 2024) | 0.33 | 0.36 | 0.30 | 0.22 |
| HEGNN (Cen et al., 2024) | 0.42 | 0.52 | 0.39 | 0.24 |
| GF-NODE (Sun et al., 2025) | 0.62 | 0.83 | 0.67 | 0.51 |
| **PAINET** | 0.20 | 0.26 | 0.21 | 0.15 |

| Model | *Walk* (S2T) | | *Run* (S2T) | |
|---|---|---|---|---|
| | Training Time (s) | Inference Time (s) | Training Time (s) | Inference Time (s) |
| EGNO (Xu et al., 2024) | 0.18 | 0.23 | 0.19 | 0.17 |
| HEGNN (Cen et al., 2024) | 0.20 | 0.25 | 0.22 | 0.21 |
| GF-NODE (Sun et al., 2025) | 1.10 | 1.26 | 0.76 | 0.68 |
| **PAINET** | 0.28 | 0.33 | 0.31 | 0.26 |

## F.7 COMPUTATION RESOURCES

All experiments are conducted on a single NVIDIA H20 GPU with 4 MiB L2 cache and 96 GiB of memory. The system is configured with NVIDIA driver version 550.144.03 and CUDA toolkit version 12.4. To evaluate the scalability of our model across datasets of varying sizes and temporal lengths, we report GPU memory usage and time cost in Fig. 7. All measurements are obtained under identical hardware and software configurations, ensuring a fair comparison across tasks.

## F.8 TIME CONSUMPTION

To evaluate the model's computational efficiency, we report both training and inference time on the Motion Capture experiments. As shown in Table 9, our model consumes comparable time costs as the powerful competitors EGNO, HEGNN and GF-NODE.

## F.9 PHYSICALLY MEANINGFUL METRIC

While our primary evaluation employs the standard Mean Squared Error (MSE) metric which is commonly adopted in the literature such as EGNO and GF-NODE, we acknowledge that MSE alone may not fully capture the physical validity of the simulation. To provide a more rigorous assessment of structural fidelity, we supplement our analysis with Root Mean Square Deviation (RMSD). Al-

Table 10: Test RMSD ($\downarrow$) for S2S ($T = 1$) on MD17.

| Model | S2S ($T = 1$) | | | |
|---|---|---|---|---|
| | **aspirin** ($\times 10^{-2}$) | **benzene** ($\times 10^{-1}$) | **ethanol** ($\times 10^{-2}$) | **malonaldehyde** ($\times 10^{-1}$) |
| Linear (Satorras et al., 2021) | $3.46 \pm 0.000$ | $4.09 \pm 0.000$ | $2.37 \pm 0.000$ | $1.46 \pm 0.000$ |
| MPNN (Gilmer et al., 2017) | $3.10 \pm 0.053$ | $2.39 \pm 0.569$ | $2.22 \pm 0.084$ | $1.14 \pm 0.004$ |
| RF (Köhler et al., 2019) | $3.30 \pm 0.003$ | $3.55 \pm 0.028$ | $2.15 \pm 0.001$ | $1.14 \pm 0.006$ |
| EGNN (Satorras et al., 2021) | $3.19 \pm 0.158$ | $2.53 \pm 0.246$ | $2.16 \pm 0.005$ | $1.13 \pm 0.001$ |
| ClofNet (Du et al., 2022) | $3.07 \pm 0.029$ | $\underline{2.19 \pm 0.010}$ | $2.15 \pm 0.006$ | $1.13 \pm 0.001$ |
| EGNO (Xu et al., 2024) | $3.07 \pm 0.122$ | $2.49 \pm 0.248$ | $2.15 \pm 0.004$ | $1.13 \pm 0.001$ |
| HEGNN (Cen et al., 2024) | $3.06 \pm 0.056$ | $2.32 \pm 0.620$ | $2.15 \pm 0.001$ | $1.14 \pm 0.002$ |
| GF-NODE (Sun et al., 2025) | $\underline{3.02 \pm 0.030}$ | $2.50 \pm 0.602$ | $2.15 \pm 0.001$ | $\underline{1.13 \pm 0.000}$ |
| **PAINET** | $\mathbf{2.99 \pm 0.040}$ | $\mathbf{2.16 \pm 0.022}$ | $\mathbf{2.08 \pm 0.004}$ | $\mathbf{1.12 \pm 0.001}$ |

| Model | S2S ($T = 1$) | | | |
|---|---|---|---|---|
| | **naphthalene** ($\times 10^{-3}$) | **salicylic** ($\times 10^{-3}$) | **toluene** ($\times 10^{-1}$) | **uracil** ($\times 10^{-3}$) |
| Linear (Satorras et al., 2021) | $2.50 \pm 0.000$ | $3.71 \pm 0.000$ | $1.11 \pm 0.000$ | $3.11 \pm 0.000$ |
| MPNN (Gilmer et al., 2017) | $2.16 \pm 0.036$ | $3.07 \pm 0.023$ | $1.03 \pm 0.042$ | $2.52 \pm 0.028$ |
| RF (Köhler et al., 2019) | $1.99 \pm 0.029$ | $3.52 \pm 0.105$ | $1.04 \pm 0.001$ | $2.49 \pm 0.050$ |
| EGNN (Satorras et al., 2021) | $1.98 \pm 0.042$ | $3.22 \pm 0.154$ | $1.03 \pm 0.015$ | $2.42 \pm 0.082$ |
| ClofNet (Du et al., 2022) | $1.97 \pm 0.066$ | $3.21 \pm 0.581$ | $\underline{1.01 \pm 0.002}$ | $2.49 \pm 0.415$ |
| EGNO (Xu et al., 2024) | $1.95 \pm 0.077$ | $3.00 \pm 0.641$ | $1.03 \pm 0.005$ | $2.35 \pm 0.149$ |
| HEGNN (Cen et al., 2024) | $\underline{1.93 \pm 0.037}$ | $\underline{2.89 \pm 0.021}$ | $1.02 \pm 0.004$ | $\underline{2.30 \pm 0.047}$ |
| GF-NODE (Sun et al., 2025) | $1.93 \pm 0.166$ | $2.93 \pm 0.254$ | $1.02 \pm 0.002$ | $2.40 \pm 0.017$ |
| **PAINET** | $\mathbf{1.83 \pm 0.008}$ | $\mathbf{2.79 \pm 0.007}$ | $\mathbf{1.00 \pm 0.000}$ | $\mathbf{2.27 \pm 0.004}$ |

Table 11: Test RMSD ($\downarrow$) for S2T ($T = 8$) on MD17.

| Model | S2T ($T = 8$) | | | |
|---|---|---|---|---|
| | **aspirin** ($\times 10^{-2}$) | **benzene** ($\times 10^{-1}$) | **ethanol** ($\times 10^{-2}$) | **malonaldehyde** ($\times 10^{-1}$) |
| EGNO (Xu et al., 2024) | $3.06 \pm 0.049$ | $2.42 \pm 0.182$ | $\underline{2.15 \pm 0.002}$ | $1.13 \pm 0.001$ |
| HEGNN (Cen et al., 2024) | $3.09 \pm 0.024$ | $2.65 \pm 0.233$ | $2.16 \pm 0.001$ | $1.14 \pm 0.001$ |
| GF-NODE (Sun et al., 2025) | $\underline{3.04 \pm 0.017}$ | $\underline{2.34 \pm 0.228}$ | $2.15 \pm 0.006$ | $\underline{1.13 \pm 0.000}$ |
| **PAINET** | $\mathbf{2.97 \pm 0.045}$ | $\mathbf{2.21 \pm 0.295}$ | $\mathbf{2.13 \pm 0.001}$ | $\mathbf{1.12 \pm 0.001}$ |

| Model | S2T ($T = 8$) | | | |
|---|---|---|---|---|
| | **naphthalene** ($\times 10^{-3}$) | **salicylic** ($\times 10^{-3}$) | **toluene** ($\times 10^{-1}$) | **uracil** ($\times 10^{-3}$) |
| EGNO (Xu et al., 2024) | $\underline{1.99 \pm 0.055}$ | $\underline{2.92 \pm 0.070}$ | $1.03 \pm 0.009$ | $\underline{2.38 \pm 0.167}$ |
| HEGNN (Cen et al., 2024) | $2.07 \pm 0.001$ | $3.00 \pm 0.024$ | $1.03 \pm 0.007$ | $2.47 \pm 0.062$ |
| GF-NODE (Sun et al., 2025) | $2.14 \pm 0.065$ | $2.93 \pm 0.009$ | $\underline{1.01 \pm 0.002}$ | $2.41 \pm 0.008$ |
| **PAINET** | $\mathbf{1.80 \pm 0.119}$ | $\mathbf{2.81 \pm 0.142}$ | $\mathbf{1.00 \pm 0.002}$ | $\mathbf{2.25 \pm 0.011}$ |

Table 12: Test RMSD ($\downarrow$) under multi-step prediction on Adk protein dynamics. All models are trained for single-step prediction and tested for five-step prediction.

| Model | S2T ($T = 5$) | | | | | |
|---|---|---|---|---|---|---|
| | **RMSD**($t = 1$) | **RMSD**($t = 2$) | **RMSD**($t = 3$) | **RMSD**($t = 4$) | **RMSD**($t = 5$) | **A-RMSD** |
| EGNO (Xu et al., 2024) | 1.058 | 1.273 | 1.380 | 1.463 | 1.497 | 1.344 |
| HEGNN (Cen et al., 2024) | 1.051 | $\underline{1.269}$ | $\underline{1.369}$ | $\underline{1.414}$ | $\underline{1.445}$ | $\underline{1.317}$ |
| GF-NODE (Sun et al., 2025) | $\underline{1.046}$ | 1.281 | 1.383 | 1.431 | 1.454 | 1.327 |
| **PAINET** | **1.037** | **1.268** | **1.324** | **1.356** | **1.412** | **1.286** |

though universally accepted accuracy thresholds for multi-body dynamics are difficult to define due to system-specific variability, we benchmark our performance against criteria established in recent literature. For protein systems, prior work suggests that an RMSD $< 2$ characterizes high-quality predictions (Xu et al., 2025). Furthermore, in molecular modeling, deviations within the range of $0.01 - 0.1$ are considered consistent with reasonable physical bond length fluctuations (Fedders & Drabold, 1996). The results in Table 10, Table 11 and Table 12 align with these standards, demonstrate that our proposed model effectively yield high-quality predictions that preserve the physical properties.

### F.10 QUALITATIVE VISUALIZATION

We provide more demonstration for the trajectories predicted by PAINET in Fig. 8, Fig. 9, Fig. 10 and Fig. 11.

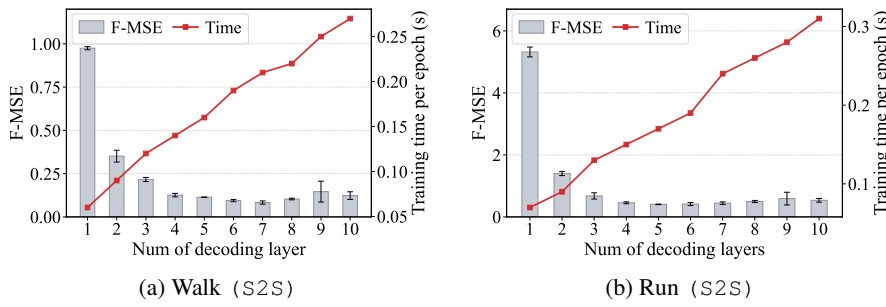

(a) Walk (S2S)

(b) Run (S2S)

Figure 5: Ablation studies w.r.t. the number of decoding layers on Motion Capture.

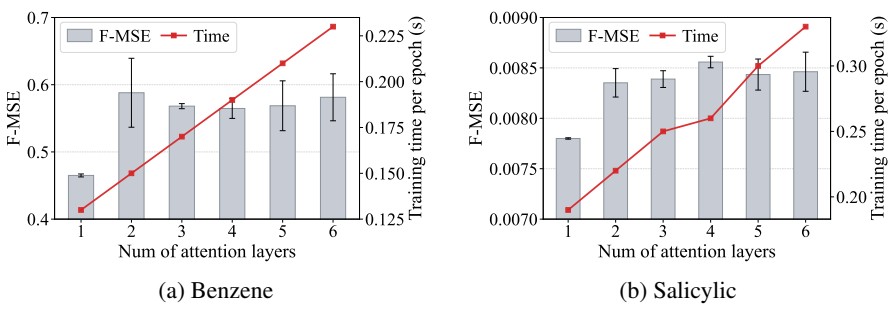

(a) Benzene

(b) Salicylic

Figure 6: Ablation studies w.r.t. the number of attention layers on Molecular Dynamics.

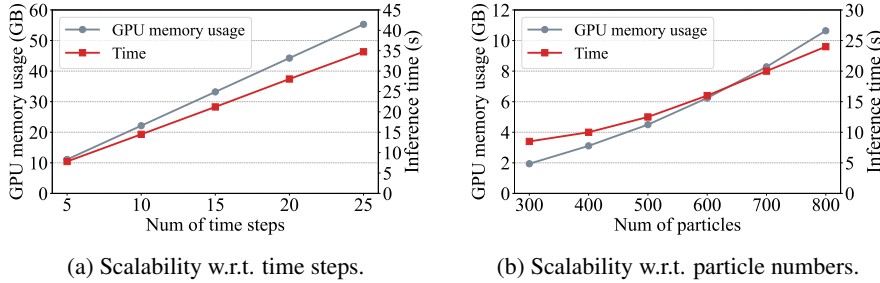

(a) Scalability w.r.t. time steps.

(b) Scalability w.r.t. particle numbers.

Figure 7: Scalability test including inference time and GPU memory cost w.r.t. time steps and particle numbers on Proteins (Adk).

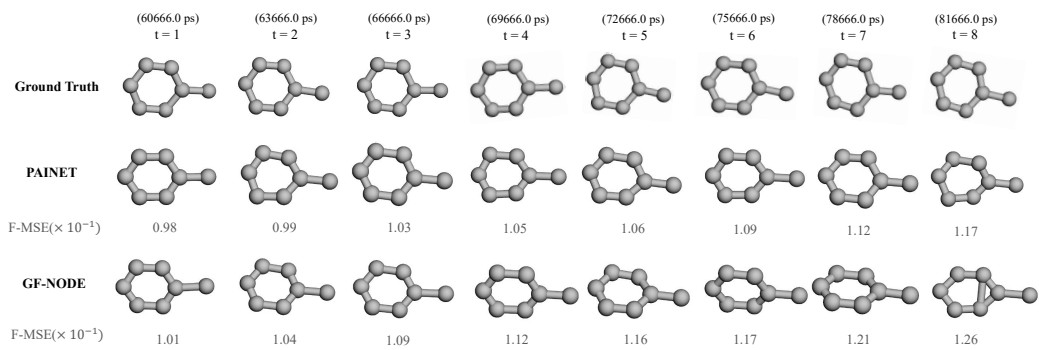

Figure 8: Representative snapshots of toluene molecular dynamics, initialized at snapshot 60666.0 ps. PAINET preserves structural characteristics and steady results in long time horizons.

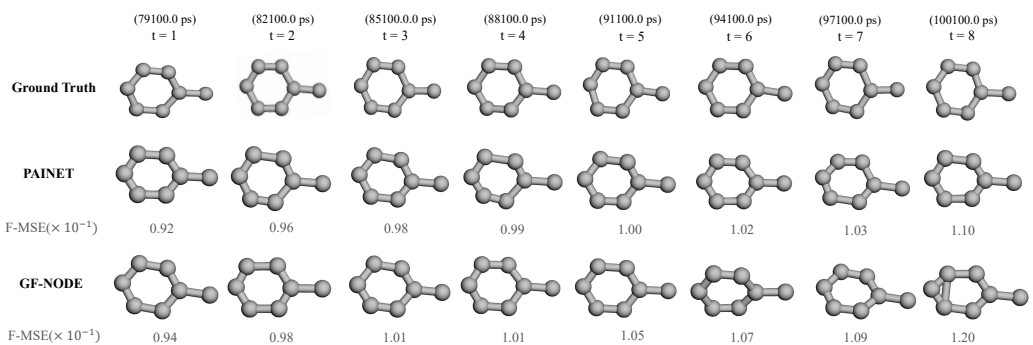

Figure 9: Representative snapshots of toluene molecular dynamics starting at 79100.0 ps.

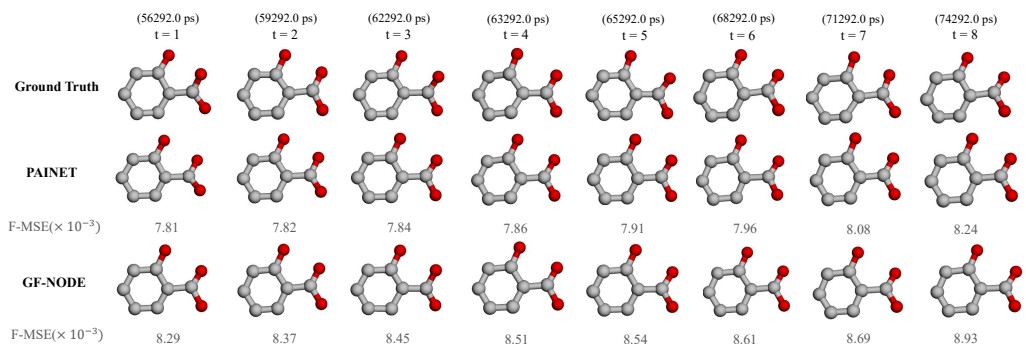

Figure 10: Representative snapshots of salicylic molecular dynamics starting at 56292.0 ps.

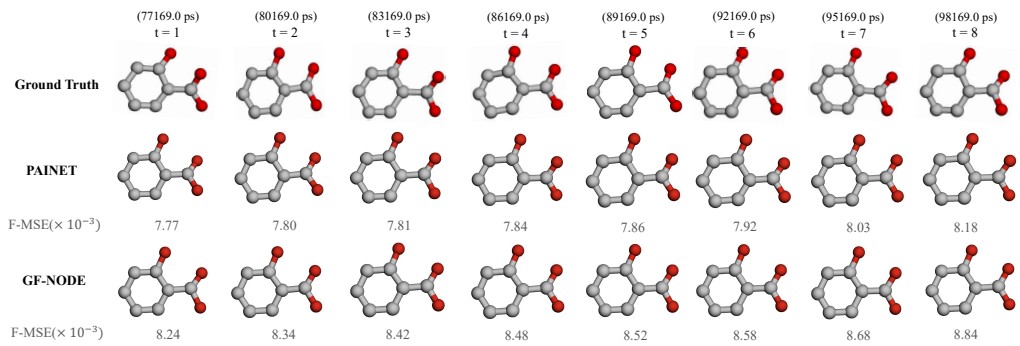

Figure 11: Representative snapshots of salicylic molecular dynamics starting at 77169.0 ps.

