# OpenReview forum: "PAINET: A Principled Efficient Transformer for 3D Dynamics Modeling"
_ICLR.cc/2026/Conference — ICLR 2026 Poster_

### Official Review · Reviewer_FRS4 · 2025-10-28

**Soundness:** 2
**Presentation:** 2
**Contribution:** 2
**Rating:** 4
**Confidence:** 5

**Summary:**

This paper addresses a key limitation of existing GNN-based methods in 3D dynamic modeling of multi-body systems: over-reliance on explicit observed structures, which fails to capture unobserved all-pair interactions critical for complex physical behaviors. It proposes PAINET, an SE(3)-equivariant neural architecture and evaluates on three real-world benchmarks: human motion capture (CMU dataset), molecular dynamics (MD17 dataset), and large-scale protein simulation (Adk dataset), comparing with classic models (Linear, RF, MPNN, EGNN) and state-of-the-art (SOTA) models (EGNO, HEGNN, GF-NODE).

**Strengths:**

1. model operation deisng with physical priors
2. several benchmark evaluations on different scenarios
3. good efficiency

**Weaknesses:**

1. Incomplete Theoretical Analysis​
Insufficient explanation of the energy function’s physical meaning: While extended from Zhou et al.’s (2004) quadratic energy, its connection to real physical systems (e.g., van der Waals forces, protein hydrogen bonds) is unclear. The concavity assumption of the ρᵢⱼ function only cites Yang et al. (2021) without analyzing its validity boundaries in 3D dynamic scenarios (e.g., particle type differences, dynamic interaction changes), raising doubts about applicability in extreme cases (e.g., strong nonlinear force-driven systems).​
Though SE(3) equivariance is fully proven, model performance under complex symmetries (e.g., mirror, scaling symmetry) common in real physics is unaddressed. Additionally, equivariance preservation of adaptive pairwise mappings Φ and Ψ in the attention network is not verified individually; while overall equivariance holds, the symmetry transfer mechanism of individual modules remains unclear.​
2. Room for Improvement in Experimental Design​
Molecular dynamics experiments ignore force field impacts (e.g., CHARMM, AMBER), only validating on MD17’s quantum force field, limiting generalization to classical force fields. Testing focuses on 8 common small molecules, lacking complex molecules (e.g., metal-containing molecules, polymers), restricting applicability in specialized chemical scenarios.​
Protein simulation relies solely on Adk protein equilibrium trajectories, excluding other protein types (e.g., membrane proteins, antibodies) or non-equilibrium dynamics (e.g., protein folding, ligand binding), insufficiently evaluating performance in complex conformational changes. Multi-step prediction only tests T=5, without exploring error accumulation at longer steps (e.g., T=10, T=20), inadequately validating long-term prediction ability.
3. More evalutions on large molecules, such as MD22.

**Questions:**

The authors should seriously address the concerns shown in Weakness point by point to improve the quality of the paper.

---

> ### Author Response · Authors · 2025-11-21
> **Rebuttal Reply to Reviewer FRS4 (1/2)**
>
> Thank you for your time and insightful reviews. We are glad that you liked our model design. We are pleased that you appreciate our model design. We have provided our responses along with additional experimental results to address your concerns:
>
> > W1.1 Insufficient explanation of the energy function’s physical meaning.
>
> The energy function (Eqn. 3) is a phenomenological model inspired by fundamental physics, not a direct substitution for a specific physical force field (like Lennard-Jones). Its connection to real systems lies in its ability to model the general principles of interaction and consistency that govern these phenomena. The term $\lambda\sum_{i,j}\rho_{ij}(||h_{i}-h_{j}||_{2}^{2})$ in the energy function acts as a generalized interaction potential in the latent space. In common systems, the particle interactions drive particles to seek stable, lower-energy configurations. The energy function formalizes this by having the model push particle embeddings ($h_i$) toward configurations that optimize internal consistency (smoothness), which indirectly reflects these binding interactions. On top of this, we draw explicit motivation from the Landau-Ginzburg form (Eqn. 6). This potential is a central phenomenological model for explaining the formation of ordered states (e.g., in superconductivity or ferro-magnetism) by characterizing the energy change associated with symmetry breaking. Our formulation uses this form to model how learned embeddings spontaneously organize (or form "latent structures") across layers. A more comprehensive explaination on the established connection between energy functions and modern graph neural networks is supplemented in **Appendix C in the revised paper** as additional technical background.
>
> > W1.2: The applicability of concave function $\rho_{ij}$ in extreme cases.
>
> The concavity assumption for $\rho_{ij}$ is primarily a mathematical requirement adopted to derive a stable and convergent optimization method, not a constraint imposed by physical force laws. As detailed in the proof for Theorem 1 (Appendix C), the concavity of $\rho_{ij}$ is essential because it enables the conversion of the non-convex energy minimization (Eqn. 3) into the iterative minimization of its variational upper bound (Eqn. 14) using convex analysis principles, leading directly to the proved convergence (Theorem 1). In Section 3.1, we instantiate $\rho_{ij}$ using a quadratic potential form derived from the Landau-Ginzburg model (Eqn. 6), where parameters $a_{ij}$ and $b_{ij}$ are constrained to ensure the required concavity and non-negativity ($a_{ij}>8b_{ij}$ and $b_{ij}>0$).
>
> In terms of the applicability, as discussed in Section 3.1 (Energy with Latent Structures), the concavity helps ensure robustness by preventing over-regularization on large differences in embeddings, which is beneficial for filtering noisy interactions. For highly nonlinear force-driven systems, while the true energy landscape is complex, our model learns a simplified, smooth descent trajectory in the latent space. The concavity constraint is imposed on the latent interaction function $\rho_{ij}$, which acts as a proxy for the actual complex forces, allowing the system to converge stably regardless of the physical force complexity. The Adaptive Pairwise Mappings ($\Phi$ and $\Psi$, defined in Section 3.2) further accommodate particle-type differences by learning specific coefficients for each pair.
>
> > W1.3: Equivariance preservation of adaptive pairwise mappings $\phi$ and $\psi$ in the attention network is not verified.
>
> Our focus on $SE(3)$ equivariance (defined in Section 2 and Appendix B) is due to its recognition as the essential geometric prior for non-relativistic physical systems and its standard use across state-of-the-art dynamics models. Extending the model to handle reflections (mirror symmetry) or scaling would require incorporating high-order geometric features or more complex Tensor Field Networks, which is reserved for future work.
>
> In terms of the equivariance preservation of our model, we explain in detail below. As defined in Section 3.2, $\Phi$ and $\Psi$ determine the pairwise coefficients $\phi_{ij}$ and $\psi_{ij}$ based only on the particle types ($z_i, z_j$) of particles $i$ and $j$ (Eqn. 9). Particle type is a discrete, intrinsic property and is thus invariant to any rigid body transformation (rotation $Q$ and translation $g$) or permutation $P$ of the coordinates. Since $\phi_{ij}$ and $\psi_{ij}$ are defined solely by these invariant features, the $\Phi$ and $\Psi$ matrices themselves, and thus the entire attention computation (Eqn. 8), are invariant with respect to $SE(3)$ transformations. The overall equivariance holds because the attention network produces invariant embeddings $H^{(t)}$ (as proven in **Appendix E: Invariance of embeddings**), which are then fed into the demonstrably $SE(3)$-equivariant decoder (EGNN) (as proven in **Appendix E: Equivariance of decoder**).

---

> ### Author Response · Authors · 2025-11-21
> **Rebuttal Reply to Reviewer FRS4 (2/2)**
>
> > W2.1: Molecular dynamics experiments that include more force fields and molecular complexity.
> > W3: More evalutions on large molecules, such as MD22.
>
> We agree that generalization across force fields and molecular complexity is paramount. We have addressed this by expanding our molecular dynamics evaluation to include the MD22 dataset [1], which features larger and more complex molecules. We focus on machine-learning interatomic potentials (MLIPs), where the target is to learn the potential energy surface (PES) and forces computed by high-level ab initio (quantum mechanical) methods (like DFT for MD17). MLIPs are force-field agnostic; they aim to model the true underlying quantum forces, not replicate empirical force fields like CHARMM or AMBER. Testing on classical force field data (which are approximations themselves) is generally not the goal of MLIP research. Our MD17 benchmark validates performance against the high-fidelity quantum force field (DFT/PBE).
>
> To address the limitation of small molecules, we add experiments on MD22. The MD22 dataset [1] includes molecules and supramolecules ranging from 42 atoms up to 370 atoms (e.g., small peptides, nanotubes), significantly exceeding the size of MD17 molecules (up to ~21 atoms). The results of PAINET as well as all baselines are reported in **Section 4.4 and Table 4 in the revised paper** which further validates the superiority of PAINET for predicting the dynamics of larger, more complex molecules.
>
> Given the complexity of molecular dynamics modeling, evaluating metal-containing compounds or polymers is a highly specialized task often requiring distinct data curation (e.g., reactive force fields) and is considered future work. Crucially, our expanded set of MD experiments (MD17 + MD22) is equal to or greater in scope than those presented by contemporary baselines like EGNO and HEGNN.
>
> > W2.2: Protein simulation experiments with more diversity and dynamics.
>
> We choose the Adk protein for evaluation as its equilibrium trajectory remains a robust and common benchmark for testing long-range structured modeling and scalability to large graphs (~3,000 atoms). This scenario primarily tests the model's ability to maintain a folded structure during complex, large-scale motion. Testing non-equilibrium processes like protein folding or ligand binding requires specialized datasets (e.g., featuring bond breaking/formation or rare-event sampling) and highly specific model extensions (e.g., reactive potentials or enhanced sampling). While important, this is a significant and focused area of research that we reserve for future work.
>
> It is worth noting that our experimental results including Motion Capture, MD17, MD22, and large-scale protein simulation have already surpassed the scope of experiments conducted by the baseline papers (EGNO, HEGNN and GF-NODE). We thus have reasonable grounds to believe that our current evaluation is sufficiently strong.
>
> > W2.3: Multi-step prediction with longer time steps
>
> In light of the suggestion, we supplement more evaluation with longer time steps with $T=10, 15, 20$ (previously $T=1, 5$) **in Section 4.6 and Table 6 in the revised paper**. PAINET consistently achieves the best performance on the Motion Capture dataset, demonstrating its strong stability in long-horizon prediction.
>
> [1] Accurate global machine learning force fields for molecules with hundreds of atoms. Science Advances, 2023

---

### Official Review · Reviewer_vf4Y · 2025-11-02

**Soundness:** 3
**Presentation:** 3
**Contribution:** 3
**Rating:** 6
**Confidence:** 4

**Summary:**

Current state-of-the-art models for 3D dynamics prediction (e.g., GNNs, EGNNs) rely on explicitly observed structures between particles. This limits their ability to capture crucial unobserved interactions (e.g., long-range forces, dynamically forming structures), leading to errors in long-term trajectory prediction and simulation, especially in complex systems like molecular dynamics and protein folding. A principled, SE(3)-equivariant neural architecture designed to model all-pair interactions, including unobserved ones.

**Strengths:**

Models Unobserved Interactions: This is its core innovation. Unlike previous GNN-based models that rely on fixed, observed structures (e.g., distance-based graphs), PAINET is designed to capture latent, all-pair interactions (e.g., long-range forces, dynamically forming bonds), which is crucial for accurate long-term dynamics prediction.

Principled, Physics-Inspired Formulation: The model is not a purely black-box architecture; it is derived from the minimization of an energy function, providing a theoretical foundation for its attention mechanism and linking it to physical principles.

**Weaknesses:**

A more clear description on the computational cost should be described. See Questions.

**Questions:**

1) Can you explain why the line in Figure 7 shows linear complexity with respect to particle numbers but not O(N^2)?
2) The training and inference time should be compared among the baseline.
3) Could you help to explain how to connect the motivated examples in the introduction with the examples in experiments? Are there prior knowledge on the all-pair interations in experiments?
4) I am not an expert in molecular dynamics and can not justify the significance of the accuracy improvement. Therefore, I will refer other reviewers' comments on it.

---

> ### Author Response · Authors · 2025-11-21
> **Rebuttal Reply to Reviewer vf4Y (1/1)**
>
> Thank you for the nice comments and valuable suggestions. We are encouraged that you appreciated our technical contributions including the problem significance, novelty, soundness and solid experiments. In the response below, we provide answers to your questions and add some experimental results in order to address the concerns:
>
> > W1 & Q1: Explain the linear scaling result in Figure 7.
>
> The empirical linear scaling observed in Figure 7b is due to that in practice the computational bottleneck is not from the all-pair attention but the decoder that needs to generate a trajectory of predicted positions. The decoder is implemented using an Equivariant GNN (EGNN) that operates on the sparse, observed graph structure $\mathcal{A}$. For large sparse graphs like the protein dynamics benchmark, the number of edges $|\mathcal{E}|$ scales near-linearly with the number of particles $N$. Therefore, the total time and memory cost empirically remains nearly linear with $N$, confirming that the results in Figure 7b are reasonable. This also demonstrates the model's good scalability to large multi-body systems.
>
> > Q2: Compare training and inference time among baselines.
>
> Thanks for suggesting this comparison that can definitely improve our work. In light of this, we report the training time per epoch and inference time cost of all the models in **Section 4.6 &Table 9 in the revised paper**. We found that our model consumes comparable time costs as the powerful competitors EGNO, HEGNN and GF-NODE. This further demonstrates the advantage of our model as it achieves significantly lower prediction error with the same level of computational costs.
>
> > Q3: Explain the connection between motivated examples in the introduction and experiment examples. Are there prior knowledge on the all-pair interations in experiments?
>
> The motivating examples directly address the core limitation PAINET solves: the failure of traditional GNNs to capture unobserved or non-local interactions. As a key motivation example, the long-range Van der Waals forces in molecules and emergent structures in protein folding highlight crucial, unobserved all-pair dependencies. This naturally inspires the model design of PAINET, i.e., the Physics-Inspired All-Pair Attention Network, which implicitly learns these unobserved dependencies (a.k.a. latent structures) by performing an energy minimization step.
>
> We do not require explicit prior physical knowledge (like force-field coefficients). Instead, the Adaptive Pairwise Mappings ($\Phi, \Psi$) learn the particle-type-specific interaction strengths necessary for modeling these forces directly from the data.
>
> > Q4: Explain the significance of the accuracy improvement.
>
> The significance of the accuracy improvements is robustly confirmed through both statistical and qualitative evidence. PAINET provides error reductions ranging from $4.7\%$ to $41.5\%$ across diverse benchmarks. Crucially, the numerical improvements are statistically significant ($p < 0.05$) in most cases, as marked by asterisks in **Table 1, 2, and 3**. Moreover, in molecular dynamics, this lower numerical error translates directly to a superior qualitative outcome (**Figure 2** and **Appendix F.8**): PAINET better preserves the essential structural characteristics (e.g., ring shapes, molecular integrity) and exhibits stability in long-time trajectories compared to competitors.

---

### Official Review · Reviewer_cVRm · 2025-11-04

**Soundness:** 3
**Presentation:** 2
**Contribution:** 2
**Rating:** 4
**Confidence:** 3

**Summary:**

This paper introduces PAINET (Physics-Inspired All-Pair Interaction Network) for 3D dynamics modeling across molecular, protein, and motion capture datasets. The method formulates an explicit energy function whose gradient defines pairwise attention weights, ensuring each network layer corresponds to an energy descent step. The encoder models implicit all-pair interactions, while the decoder uses an equivariant EGNN for SE(3)-consistent trajectory prediction. The authors claim PAINET learns latent long-range interactions without explicit graph structures, outperforming prior GNN-based models such as EGNN, HEGNN, EGNO, and GF-NODE.

**Strengths:**

1. The attention update is rigorously derived from an energy descent principle, providing a clear physical interpretation. The energy-derived attention mechanism replaces softmax with gradient-based weights, improving interpretability.
2. Experiments show consistent gains across motion, molecular, and protein dynamics tasks at comparable computational cost.
3. The paper is well-organized and readable, with clear motivation and intuitive explanations.

**Weaknesses:**

1. The reported results for key baselines, particularly EGNO and GF-NODE, appear inconsistent with their original publications, casting doubt on the validity of the claimed improvements. For example, in Table 1, PAINET reports EGNO achieving F-MSE = 14.2 (×$10^{-2}$) on Walk and F-MSE = 4.15 (×10) on Run—values that deviate substantially from EGNO’s original paper (F-MSE = 8.1 (×10-2) on Walk and F-MSE = 3.39 (×$10^{-1}$) on Run). Similar inconsistencies appear for GF-NODE and other baselines, likely due to preprocessing or implementation differences. These discrepancies need careful clarification.
2. The paper lacks ablations isolating the contributions of the energy function, Φ/Ψ mappings, and all-pair interactions. It remains unclear whether performance gains stem from the proposed energy mechanism or general architectural effects.

Missing related work and baselines:
1. SE(3) Equivariant Graph Neural Networks with Complete Local Frames; ICML 2022;
2. AlphaNet: Scaling Up Local Frame-based Atomistic Foundation Model, Npj Comput. Mater. (2025)

**Questions:**

1. Please clarify the differences in preprocessing, subject selection, frame rate, or hyperparameters that might explain the inconsistent EGNO and GF-NODE results.
2. How sensitive is the model to the specific choice of potential function ρ? Did you test other learnable monotonic forms beyond the quadratic–quartic setup?
3. Could you include or discuss comparisons removing the energy weighting, Φ/Ψ mappings, or limiting interactions to local neighbors to substantiate the claimed benefits?

---

> ### Author Response · Authors · 2025-11-21
> **Rebuttal Reply to Reviewer cVRm (1/1)**
>
> Thank you for the time and thorough reviews. We are glad that you liked our method, theory and experiments. Here are our responses to your questions:
>
> > W1 & Q1: The inconsistency of EGNO and GF-NODE results.
>
> Thank you for carefully checking the reported numbers. The inconsistent results commonly happen as reproduced by other papers, which is likely due to different random seeds in multiple runs, hardware, and software environments [1]. In our experiments, we strictly followed the settings adopted by the EGNO paper (timestep, frame rate, initial seed, and so on). However, since we reported the average results over 5 runs with different seeds, the averaged values may differ from EGNO paper. In addition, all of our experiments were conducted on NVIDIA H20 GPUs instead of NVIDIA L40 GPUs used by GF-NODE and the software environments (e.g., NumPy versions) slightly differ as well, which may also cause the inconsistency. Thus, we can see EGNO’s performance on MD17 Benzene (52.53) reported in the GF-NODE paper differs from the original EGNO paper (48.85) as well.
>
> We fully understand the reviewer's concern about the reliability of our reported results. To ensure it, all results were averaged over 5 independent runs under the same environment. We also provide our code through the anonymous link in Abstract as well as the full set of hyperparameters and implementation details in **Appendix E.5** for reproducibility.
>
> > W2 & Q3: Ablations on energy function, $\phi$/$\psi$ mappings, and all-pair interactions.
> > Q2: The sensitivity test about potential function $\rho$.
>
> Thank you for suggesting these new experiments. However, we respectively clarify that in our original submission, we have included the ablation stuides on $\phi$/$\psi$ mappings and all-pair interactions in **Figure 3*. As shown in Figure 3a, compared with the four variants using different fixed $\phi$ and $\psi$ as well as the variant "w/o attention", our model achieves superior results, validating the effectiveness of our attention network.
>
> To strengthen our ablation studies, we added comparison with more variants shown in the **updated Figure 3a in the revised paper**. For ablation study on all-pair interactions, we replace the global attention in our model with the "local attention" (that only attends on neighbored nodes), and we found the A-MSE on Motion Capture Run increases from $0.332 \pm 0.012$ to $0.568 \pm 0.013$. This further validates the efficacy of our all-pair attention. In terms of the sensitivity w.r.t. potential function $\rho$, we add comparison with two another variants: 1) "scaled quadratic $\rho$" which multiple the quadratic term of $\rho$ in Equation 6 by two; 2) "linear $\rho$" which simplifies $\rho$ as a linear functional form. The results in **updated Figure 3a** show that the scaled quadratic variant leads to slight performance drop while using linear $\rho$ degrades the performance dramatically. This validates the superiority of the quadratic form for $\rho$ adopted in our model.
>
> > W3: Missing related work and baselines.
>
> We appreciate the reviewer suggesting the related works [2, 3]. The model proposed by [3] is designed for molecular property prediction which is a different problem from what we study, i.e., 3D dynamics prediction. While [2] studies the same problems as us, it proposes a projection-based method for preserving equivalence, which is orthogonal to our model from the architectural view. That said, we have included both papers as references in the revised paper. Furthermore, we include ClofNet [2] as an additional baseline throughout all the cases on MD17 (see **Table 1** and **Table 2** in the revised paper). Our model consistently outperforms ClofNet by a clear margin. This new comparison definitely improves our empirical contribution.
>
> [1] Chen, Boyuan, et al. "Towards training reproducible deep learning models." Proceedings of the 44th international conference on software engineering. 2022.
>
> [2] SE(3) Equivariant Graph Neural Networks with Complete Local Frames, ICML 2022.
>
> [3] AlphaNet: Scaling Up Local Frame-based Atomistic Foundation Model, Npj Comput. Mater. (2025).

---

### Official Review · Reviewer_sMNn · 2025-11-04

**Soundness:** 2
**Presentation:** 1
**Contribution:** 2
**Rating:** 2
**Confidence:** 4

**Summary:**

This paper proposes PAINET, a principled SE(3)-equivariant architecture for modeling all-pair interactions in multi-body dynamical systems. The model comprises: (1) a novel physics-inspired attention mechanism derived from the minimization trajectory of an energy function, and (2) a parallel decoder designed to maintain equivariance while enabling efficient inference. Empirical evaluation across diverse real-world benchmarks demonstrates the effectiveness of the proposed approach.

**Strengths:**

* The integration of all-pair interactions with a parallel, equivariant decoder is technically well-founded and appears to be effectively implemented.
* PAINET demonstrates strong empirical performance, outperforming existing baselines on several challenging multi-body system benchmarks.

**Weaknesses:**

* Several key aspects of the methodology are insufficiently motivated or explained, making it difficult to follow.
  * The rationale for regularizing the distance between updated and current node embeddings (lines 184-185) is unclear. What specific issue does this address, and how was the regularization strength chosen?
  * The description of the underlying physical principle (lines 192-199) is presented without a clear connection to the model's technical design. It is not evident how this principle directly informs the architecture or contributes to the reported performance.
  * The introduction of the functions $\phi_{ij}$ (line 211 and Eq. 7) and  $\psi_{ij}$ (Eq. 7) appears abrupt. The logical flow and the specific roles these functions play in the overall framework need to be more clearly articulated.

* The paper lacks a well-defined, physically meaningful accuracy threshold for the multi-body dynamics tasks. Without such a benchmark, it is difficult to assess whether the reported performance improvements translate to practical utility in real-world applications.

**Questions:**

* How was the number of trajectory steps $T$ determined for different S2T tasks (e.g., $T=5$ in Table 1)? Was this hyperparameter tuned on a validation set, and is it consistent across systems with different dynamical properties?
* For predicting trajectories longer than the predefined $T$, what is the inference procedure? Does the model operate autoregressively, and if so, how are potential error accumulations mitigated?

**Details Of Ethics Concerns:**

No ethics concerns observed.

---

> ### Author Response · Authors · 2025-11-21
> **Rebuttal Reply to Reviewer sMNn (1/2)**
>
> Thank you for the time and thorough reviews. In the following response, we answer your questions point-by-point, clarify our technical details and supplement new experiments as suggested to further strengthen our contributions.
>
> > W1: Several key aspects of the methodology are insufficiently motivated or explained, making it difficult to follow.
>
> We clarify the three points raised by the reviewer below and also updated the paper accordingly (**Sections 3.1 and 3.2**) to improve the clarification.
>
> 1. **Rationale for Regularization Term**
>
>     The term $\sum_{i} || h_{i}-h_{i}^{(t)} ||^2$ in Equation 3 serves as a regularization term that prevents trivial solution and ensures smooth evolution. Firstly, it ensures the updated embeddings $h_i$ do not collapse into a single point to minimize the all-pair interaction term $\lambda\sum_{i,j}\rho_{ij}(||h_{i}-h_{j}||_{2}^{2})$. Secondly, it limits the distance of the update per step, forcing the learned latent structures to evolve smoothly and incrementally from the current state $H^{(t)}$, analogous to a small time step in physical dynamics. The regularization strength is controlled by $\lambda$ (Eqn. 3) which is empirically tuned on the validation set.
>
> 2. **Connection of Physics Principle to Technical Design**
>
>     The principle that physical systems evolve by minimizing energy directly informs the attention layer's derivation. We hypothesize that learning the optimal latent interactions is equivalent to driving particle embeddings toward a state that minimizes the energy function $E$. Theorem 1 formally proves that the update rule derived from the gradient descent of the energy $E$ results in the proposed attention update (Equation 4/5). Thus, our "Principled Attention Layer" is explicitly designed as a rigorous energy descent step in the latent embedding space. This ensures the attention mechanism is physically grounded, which is our key novelty. More explanations on the established connection between energy functions and modern graph neural networks are supplemented in **Appendix C in the revised paper** which compensates our technical background.
>
> 3. **Role of $\Phi$ and $\Psi$ (Adaptive Pairwise Mappings)**
>
>     The matrices $\Phi=[\phi_{ij}]$ and $\Psi=[\psi_{ij}]$ are the learnable, particle-type-specific coefficients for the attention weights. Conceptually, they are motivated by the observation that long-range interactions (like Van der Waals potentials) vary based on the types of the interacting particles (e.g., carbon-carbon vs. carbon-hydrogen). To implement this idea, in our model, these two matrices are designed to transform the attention equation (Eqn. 7) to weight contributions based on particle type. They are computed via look-up embeddings ($E_{\phi}, E_{\psi}$) specific to the one-hot particle types $Z$ (Eqn. 9). This allows PAINET to learn fine-grained, long-range interactions beyond the observed local structure.
>
> > W2: The paper lacks a well-defined, physically meaningful accuracy threshold for the multi-body dynamics tasks. Without such a benchmark, it is difficult to assess whether the reported performance improvements translate to practical utility in real-world applications.
>
> Our original experiments use the well-established evaluation metric for dynamics prediction (i.e., MSE) that is commonly used in the liteature including the baselines we compared (e.g., EGNO and GF-NODE). We totally agree that adding more metrics, especially the physically meaningful metrics, can significantly improve the evaluation. That said, there is no universally acknowledged accuracy threshold for multi-body dynamics because it depends heavily on the specific physical system and the properties.
>
> To further improve our evaluation, particularly regarding the practical utility of the predicted results, we supplement new evaluation on a physically meaningful metric Root Mean Square Deviation (RMSD) suggested by the recent works [1, 2]. The prior study [1] indicates that RMSD < 2 means high-quality predictions for protein systems; and for molecular dynamics, the prediction results are consistent with the bond length fluctuations if RMSD is within the range of 0.01-0.1 [2]. The results on MD17 S2S, MD17 S2T and Adk protein dynamics are presented in **Table 10-12 and Appendix F.8 of the revised paper**. Across nine different datasets, our model consistently achieves the superior RMSD compared to the baselines, which shows that the prediction of PAINET preserves the desired physical properties in particle dynamics.
>
>
> [1] Xu, Kai, et al. "Efficient Generation of Protein and Protein–Protein Complex Dynamics via SE (3)-Parameterized Diffusion Models." Journal of Chemical Information and Modeling (2025).
>
> [2] Fedders, P. A., and D. A. Drabold. "Molecular-dynamics investigations of conformational fluctuations and low-energy vibrational excitations in a-Si: H." Physical Review B 53.7 (1996): 3841.

---

> ### Author Response · Authors · 2025-11-21
> **Rebuttal Reply to Reviewer sMNn (2/2)**
>
> > Q1: How was the number of trajectory steps $T$ determined for different S2T tasks?
>
> The number of trajectory steps $T$ for S2T tasks was adopted from established evaluation protocols in the literature (e.g., EGNO) for fair comparison, not determined by hyperparameter tuning. In our original submission, we have detailed these settings in **Appendix F.2** ("For both training and evaluation, we set T=5 in S2T tasks") and **Appendix F.3** ("For both training and evaluation, we set T=8 in S2T tasks"). Therefore, $T$ is not a tuned hyperparameter but a fixed variable chosen to match domain-specific evaluation settings, ensuring consistency with competitors across varying dynamical properties.
>
>
> > Q2: For predicting trajectories longer than the predefined, what is the inference procedure? Does the model operate autoregressively, and if so, how are potential error accumulations mitigated?
>
> PAINET utilizes a recurrent-parallel inference architecture which is fundamentally **non-autoregressive** in dynamics prediction, mitigating the error accumulation suffered from autoregressive models. We have described the detailed inference procedure of our model in **Section 3.2** and **Algorithm 1** in the original submission. Specifically, the attention network computes the embeddings $\{H^{(t)}\}_{t=1}^{T}$ recurrently based on previous embeddings $H^{(t-1)}$. The equivariant decoder generates the predicted positions $\hat{X}^{(t)}$ at each time step $t$ from the learned embedding $H^{(t)}$ and, crucially, the initial position $X^{(0)}$ (ground truth), not the previous predicted position $\hat{X}^{(t-1)}$. Therefore, the model avoids the recurrent prediction of autoregressive models that propagates accumulated errors in $\hat{X}^{(t-1)}$. This non-autoregressive approach, combined with the SE(3)-equivariant geometric inductive bias, enables better generalization to longer time steps (as validated in **Table 4**). Additionally, we supplement more evaluation on the model performance for prediction in longer time steps in **Section 4.6 & Table 6**.

---

> > ### Comment · Reviewer_sMNn · 2025-11-28
> >
> > Thank the authors for the further responses, which have resolved most of my concerns. Accordingly, I will be raising the rating to 6.

---

### Author Response · Authors · 2025-11-21

Dear Area Chair and Reviewers,

We'd like to express our sincere gratitude to all the reviewers for their thorough reviews and constructive feedback. In response to the comments, we have supplemented an extensive set of new results and significantly improved the paper accordingly. These new results are highlighted in the individual response below and also incorporated into the revised paper. As a quick overview, here we summarize the updates we made for the paper:

**1. Expanded experimental scope (new datasets & baselines)**: We expanded our evaluation to a new dataset (MD22) for larger molecules **(Appendix F.11)**.
Additionally, we added a new baseline (ClofNet) for our main results **(Tables 1 & 2)**.

**2. Robustness for longer trajectories**: We added evaluation on prediction in longer time steps with $T=10, 15, 20$ steps (previously $T=1, 5$) **(Appendix F.6)**.

**3. More ablation studies:** We added ablation studies on all-pair interaction learning and potential function $\rho$ **(Figure 4)**.

**4. Physically meaningful metrics:** To address concerns regarding the practical utility, we added evaluation on Root Mean Square Deviation (RMSD), a new metric suggested by recent works that measures the structural quality and fidelity of prediction **(Appendix F.10)**.

**5. Computational efficiency:** We added comparison of training and inference time costs with baselines **(Appendix F.9)**.

**6. Clarifications:** We revised **Sections 3.1 and 3.2** and added **Appendix C** to clarify the connection between energy function and our model design.

We thank the reviewers again for the nice suggestions that significantly improve our paper, and are happy to address any further feedback or comment during the discussion.

---

### Author Response · Authors · 2025-11-25

Dear Reviewers,

We would like to express our gratitude to all the thoughtful and constructive feedback. As the author-reviewer discussion phase unfolds, we sincerely look forward to your further insights and re-evaluation of our submission.

In our response below, we have carefully addressed all the weaknesses (W) and questions (Q), and have additionally included new experimental results incorporated into the revised version of the paper. We have also made corresponding updates to the manuscript based on your valuable comments.

We greatly appreciate your time and effort in reviewing our work and look forward to any further clarification you might need.

---

### Meta-Review · Area_Chair_3AQg · 2026-01-04

**Summary:**

Summary:
The paper’s main contribution is the integration of a physically inspired mechanism into an existing equivariant graph neural network framework to improve predictive performance. Throughout the rebuttal, reviewers showed sustained interest in the origin of this mechanism, its architectural role, and the strength of the empirical evidence. The authors provided clear responses and sufficient experimental support, and I recommend acceptance.

Strength:

1. This paper provides ample empirical validation (including multi-molecule, multi-molecule scale tasks) and ablation experiments.
2. The rebuttal provides a thorough and clear explanation of the underlying physical principles.

Weakness:

Several revisions would strengthen the paper. The presentation is currently dense, and allocating more space to the physical motivation and its empirical validation would improve clarity. Standard model components such as EGNN formulas could be moved to the appendix. Reordering Thm. 1 to appear directly after Eq. (3), with appropriate cross-references to appendix proofs, would also enhance readability. Adding several visualizations to the experimental section would further clarify the intuition behind the proposed mechanism.

The descriptions of the MD17 and protein-dynamics experiments can be streamlined, with most details moved to the appendix. This would create space for a clearer and more prominent discussion of MD22, which is newly proposed in this work and therefore deserves focused exposition. These experiments could also be combined into a unified section, since the method naturally generalizes from small- to large-scale geometric graphs without requiring task-specific design. It is also important to clarify that prior MD22 benchmarks have focused exclusively on energy and force prediction, and no position-prediction task has existed in the standard MD22 setup.

The subsection containing the ablation and robustness analyses is well placed, but the title “Future Discussions” is too broad and should be renamed to “Ablation Studies”. In particular, the sensitivity analysis of the potential function — which directly addresses Reviewer cVRm’s concern — should be highlighted explicitly as a key component of the ablation rather than being merged into general robustness checks.

Finally, several benchmark citations should be corrected. MD17 (from GMN) and protein-dynamics benchmarks (from EGHN) should be attributed to their original sources rather than to later works such as EGNO that reused these datasets. Many now have formal publications, so arXiv citations should be updated accordingly.

**Reviewer Concerns:**

Most substantive concerns, particularly those related to physical motivation and the underlying mechanisms, have now been adequately addressed. The remaining points, such as those from Reviewer FRS4, appear to reflect an overly stringent standard and offer limited actionable feedback; consequently, they do not materially affect my assessment of the paper's technical merit or overall contribution. For these reasons, I have placed limited weight on that review (notwithstanding the confidence score of 5).

**Reviewer Scores:**

As the authors’ response resolves the fundamental issues, the reviewers are positioned to assign a positive rating. I anticipate an 8 from Reviewer vf4Y, a 6 from Reviewer cVRm, and a positive rating from Reviewer FRS4 once the specific misunderstanding is clarified.

---

### Decision · Program_Chairs · 2026-01-26

Accept (Poster)